# Structural basis for ryanodine receptor type 2 leak in heart failure and arrhythmogenic disorders

Marco C. Miotto [1,2] ✉, Steven Reiken[1,2], Anetta Wronska [1,2], Qi Yuan[1,2], Haikel Dridi [1,2], Yang Liu[1,2], Gunnar Weninger [1,2], Carl Tchagou[1,2] & Andrew R. Marks [1,2] ✉

Heart failure, the leading cause of mortality and morbidity in the developed world, is characterized by cardiac ryanodine receptor 2 channels that are hyperphosphorylated, oxidized, and depleted of the stabilizing subunit calstabin-2. This results in a diastolic sarcoplasmic reticulum $Ca^{2+}$ leak that impairs cardiac contractility and triggers arrhythmias. Genetic mutations in ryanodine receptor 2 can also cause $Ca^{2+}$ leak, leading to arrhythmias and sudden cardiac death. Here, we solved the cryogenic electron microscopy structures of ryanodine receptor 2 variants linked either to heart failure or inherited sudden cardiac death. All are in the primed state, part way between closed and open. Binding of Rycal drugs to ryanodine receptor 2 channels reverts the primed state back towards the closed state, decreasing $Ca^{2+}$ leak, improving cardiac function, and preventing arrhythmias. We propose a structural-physiological mechanism whereby the ryanodine receptor 2 channel primed state underlies the arrhythmias in heart failure and arrhythmogenic disorders.

Ryanodine receptor type 2 (RyR2), an intracellular $Ca^{2+}$ release channel located on the sarcoplasmic reticulum (SR) of cardiomyocytes, is essential for cardiac muscle excitation-contraction coupling[1,2]. Dysregulation of $Ca^{2+}$ handling, characterized by aberrant SR $Ca^{2+}$ leak, is a pivotal pathophysiological mechanism in a spectrum of cardiac disorders, including heart failure (HF), atrial fibrillation, catecholaminergic polymorphic ventricular tachycardia (CPVT), and other arrhythmogenic disorders[3–6]. CPVT, a rare inheritable arrhythmia syndrome that results in sudden cardiac death, has been linked to pathogenic mutations in the RyR2 gene[7,8]. Many of these mutations encode a leaky RyR2 channel, predisposed to spontaneous $Ca^{2+}$ release, particularly under conditions of heightened catecholaminergic activity, such that exercise and/or emotional stress may result in fatal cardiac arrhythmias due to RyR2-related $Ca^{2+}$ leak[4,7–9]. Until recently, the structural basis underlying this CPVT-causing leak was unknown. Structurally, RyR2 is a 2.3 MDa homotetramer consisting of

two functional regions: a massive cytoplasmic region, also known as the cytoplasmic shell, which comprises several regulatory domains (NTDs, NSol, SPRYs, RYs, JSol, and BSol domains) and the stabilizing subunit calstabin-2, and an activation core and pore region, which comprises the $Ca^{2+}$ binding site and the transmembrane domains (Supplementary Table 1). We have recently used cryo-EM to show that a PKA-phosphorylated CPVT-related RyR2-R2474S mutant channel adopts a pathological "primed" state which can be readily activated by stress signals resulting in a pathological leak of SR $Ca^{2+}$ that can trigger arrhythmias[10]. The primed state is characterized by the downward-outward shift of the cytoplasmic shell to an intermediate conformation between the closed and open states and is reversed by binding of the Rycal ARM210 to the RY1&2 domain or binding of calmodulin to the BSol and SCLP domains[10].

RyR2-mediated diastolic SR $Ca^{2+}$ leak has also been implicated in HF and atrial fibrillation[3,4]. In HF, a complex syndrome characterized by

[1]Department of Physiology and Cellular Biophysics, Columbia University Vagelos College of Physicians and Surgeons, New York, NY, USA. [2]Clyde and Helen Wu Center for Molecular Cardiology, Columbia University, New York, NY, USA. ✉e-mail: mm5642@cumc.columbia.edu; arm42@cumc.columbia.edu

reduced cardiac output, RyR2 remodeling due to post-translational modifications results in diastolic SR $Ca^{2+}$ leak, culminating in diminished SR $Ca^{2+}$ content, reduced systolic $Ca^{2+}$ release, and impaired myocardial contractility. This diastolic SR $Ca^{2+}$ leak can also trigger arrhythmogenic events, further impairing cardiac performance, and ultimately resulting in cardiac arrest, the leading cause of death in HF patients[11,12]. In failing hearts, RyR2 is PKA-hyperphosphorylated, oxidized, nitrosylated, and depleted of the stabilizing subunit calstabin-2, resulting in leaky channels that promote cardiac dysfunction and arrhythmias[13–17]. If untreated, patients eventually develop organ failure, cognitive dysfunction[18], arrythmias, and cardiac arrest[14]. Atrial fibrillation, the most common cardiac arrhythmogenic disorder, occurs when the atria undergo uncoordinated rapid contractions causing the atria to fibrillate, resulting in a decreased cardiac output[19]. Atrial fibrillation can increase the risk of blood clots in the heart leading to embolic strokes[20]. While HF and atrial fibrillation are complex and multi-factorial diseases, there is strong evidence that they share common elements and that diastolic SR $Ca^{2+}$ leak via RyR2 plays an important role in the progression of both disorders[3,4]. Common elements are diastolic SR $Ca^{2+}$ leak, oxidative stress, mitochondrial dysfunction[21,22], and leaky RyR2 characterized by PKA hyperphosphorylation, calstabin depletion, oxidation, nitrosylation, and glutathionylation[3,23]. Validating these observations, a mouse model carrying the phosphomimetic mutation RyR2-S2808D develops HF[16] and a knockout of calstabin-2 facilitates atrial fibrillation[24]. Interestingly, a mouse model overexpressing calstabin-2 rescued HF[25] and three mouse models carrying CPVT mutations exhibit atrial fibrillation[26], supporting a shared mechanism amongst these cardiac disorders.

Since HF patients exhibit both diastolic SR $Ca^{2+}$ leak and reduced systolic $Ca^{2+}$ release, we decided to study two accessory proteins that play important roles in diastolic and systolic RyR2 regulation: calstabin-2 (also known as FKBP1B or FKBP12.6) and calmodulin (CaM), respectively[27]. Calstabin-2 is a RyR2 closed state stabilizer constitutively bound to RyR2 in healthy conditions, but depleted in disease conditions[26,28,29]. Since calstabin-2 stabilizes the closed state of RyR2, it has a major regulatory role during diastole (when the heart is relaxed, cytosolic $Ca^{2+}$ concentrations remain in the nanomolar range, and RyR2 remains in the closed state). CaM is a $Ca^{2+}$ binding protein which, in the $Ca^{2+}$ bound form ($Ca^{2+}$-CaM), binds strongly to and inhibits RyR2, suggesting it has a role in the regulation and termination of SR $Ca^{2+}$ release, and deactivation of RyR2[30,31]. Since CaM regulation is $Ca^{2+}$-dependent, it has a major regulatory role during systole (when the heart is contracting due to increased $[Ca^{2+}]_{cyt}$). Moreover, oxidation of CaM reduces its affinity for RyR2, suggesting CaM inhibition is diminished in the pathological oxidative conditions found in HF[32,33].

In order to determine the structural basis for the RyR2-mediated diastolic SR $Ca^{2+}$ leak observed in arrhythmic and failing hearts we solved the structures of mutant RyR2 channels that can cause CPVT or HF, including the phosphomimetic RyR2-S2808D mutant channel which results in a cardiomyopathy with reduced cardiac function[16] and the CPVT mutant channels RyR2-R420Q and RyR2-R420W[34–41]. We reasoned that these disease-causing mutant forms of RyR2 would provide insights regarding the structural basis of the diastolic SR $Ca^{2+}$ leak that promotes HF and fatal cardiac arrhythmias and potentially serve as models to test new treatments for these common disorders. Here, we present the cryo-EM structures of the post-translationally modified RyR2-S2808D in the presence of low non-activating $Ca^{2+}$, conditions that resemble those of RyR2 in failing hearts during diastole. We now show that the mutant RyR2 channel is present in a "subprimed" state (a state between the closed and primed states which can be readily and inappropriately activated, resulting in SR $Ca^{2+}$ leak), that the Rycal drug ARM210 can stabilize the closed state, and that depletion of calstabin-2 (as occurs in failing hearts) induces a primed state, like that observed in the mutant CPVT channels. We also present the cryo-EM structures of the PKA-phosphorylated human CPVT-linked RyR2-R420Q and RyR2-R420W under conditions that resemble those present during diastole and show that both mutant channels are in the primed state and that the Rycal drug ARM210 stabilizes the closed state of the mutant channels. We further analyze the affinity of calstabin-2 for the different CPVT-linked mutant channels. The affinity of calstabin-2 for RyR2 correlates with the degree of structural changes introduced by the mutations (i.e. the primed state) and the age of onset of symptoms of the CPVT patients, suggesting that the affinity of calstabin-2 for RyR2 is a predictive tool of the level of primed state and the pathogenicity of remodeled RyR2. Furthermore, we present the cryo-EM structures of RyR2-R420W, under conditions that resemble those present during systole, in the presence and absence of CaM, which also exhibits stabilizing effects on the mutant channels. In conclusion, the present study supports our hypothesis that the pathological RyR2 primed state is a key common mechanism underlying diastolic SR $Ca^{2+}$ leak that leads both to the arrhythmic events in HF and arrhythmogenic disorders like CPVT and atrial fibrillation, and to the reduction in cardiac function observed in HF patients caused by depleting the SR $Ca^{2+}$ store.

## Results

### Heart failure RyR2-S2808D channels are in the primed state

We previously showed that RyR2-S2808 is the main target of PKA phosphorylation[10] and that pS2808 likely interacts with residues R1500 and K1525 and stabilizes the RY3&4 domain inducing the subprimed state (Supplementary Fig. 1a,b). In support of this hypothesis, we now show that neutralization of the positively charged residues R1500 and K1525 blocks the increased open probability of RyR2 under cAMP-induced PKA activation, confirming our previous hypothesis and the importance of the PKA site at RyR2-S2808 (Supplementary Fig. 1c-e)[13,15]. RyR2-S2808D channels were recombinantly expressed in HEK-293 cells and isolated as previously reported[10]. In addition to the phosphomimetic RyR2-S2808D mutation, the HEK-293 cells were treated with $H_2O_2$ to induce oxidative modifications of RyR2 that are a hallmark of HF[3]. In WT RyR2 expressing cells, $H_2O_2$ treatment results mostly in oxidation with a low level of nitrosylation of RyR2 (Supplementary Fig. 1f). In the RyR2-S2808D expressing cells, we observed a basal level of oxidation and nitrosylation which is exacerbated by treatment with $H_2O_2$ (Supplementary Fig. 1f). This suggests that chronic phosphorylation of S2808 results in $Ca^{2+}$ leak-induced oxidative stress that leads to oxidation and nitrosylation of RyR2 resulting in a vicious cycle leading to more $Ca^{2+}$ leak and further oxidative stress. The increase in nitrosylation could be the consequence of the impaired nitric oxide clearance due to the redox imbalance[42]. To avoid RyR2 aggregation during the purification, we used reducing agents (DTT and TCEP), which can reverse most reversible oxidation and nitrosylation modifications, obtaining purified RyR2 with irreversible oxidative modifications (Supplementary Fig. 1g). Irreversible oxidation is also a hallmark of HF which is detected by mass spectrometry and DNP-based assays used in previous studies[22,43,44].

We determined the cryo-EM structures of the RyR2-S2808D channels under conditions that resemble diastole in the heart when cardiomyocytes are at rest (concentration of free cytosolic $Ca^{2+}$ below 150 nM and ATP in the mM range, Supplementary Figs. 2a, b, 3a, Supplementary Table 2). Under these conditions, we found that the pore of the RyR2-S2808D channels with irreversible oxidative modifications remained closed (Supplementary Fig. 4), but the cytoplasmic shell showed a downward-outward movement compared to the closed WT RyR2, similar to that of the PKA-phosphorylated RyR2 (Fig. 1a, b)[10]. Analysis of the root mean squared deviation (RMSD) normalized projection showed that the architecture of the RyR2-S2808D channel was between the closed and the pathological primed states, in a "subprimed" state (RMSD = 0.11, Fig. 1c). The subprimed state would lower the activation barrier of the channel facilitating SR $Ca^{2+}$ leak. We were

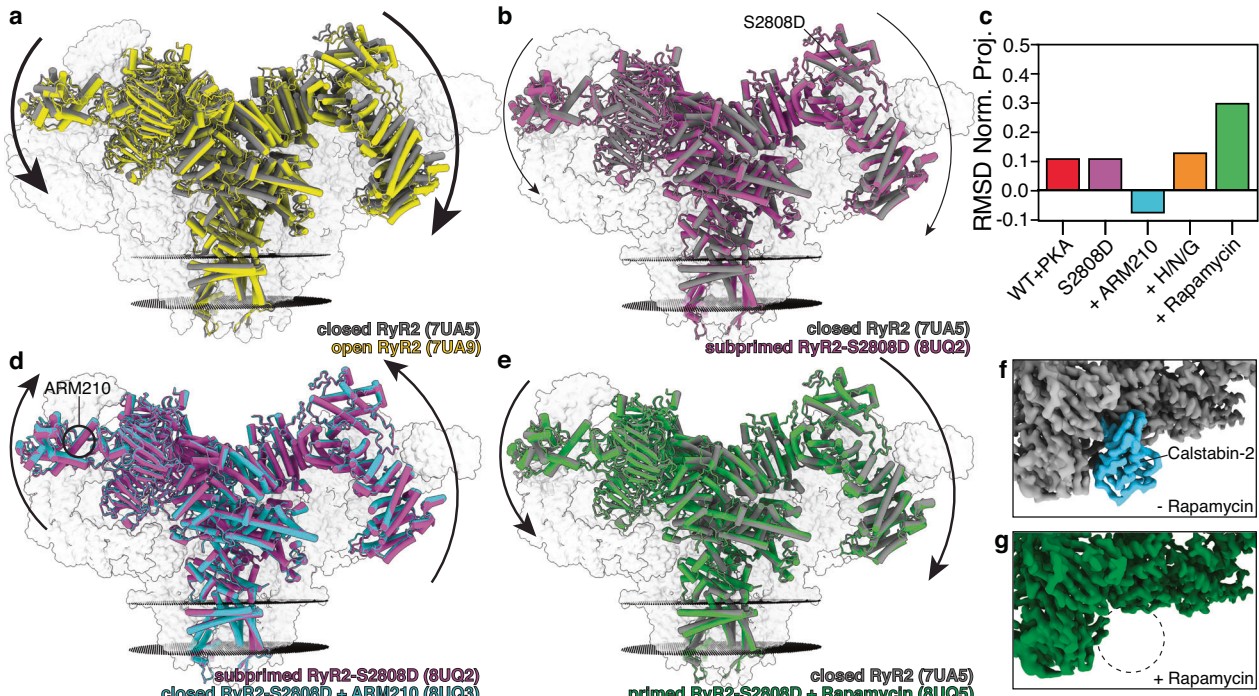

**Fig. 1 | Cryo-EM reconstructions of human RyR2-S2808D: Rycal ARM210 stabilizes the closed state while calstabin-2 depletion induces the primed state. a** Overlapped models of closed RyR2 (PDB:7UA5, gray) and open RyR2 (PDB:7UA9, yellow). The arrows show that the cytoplasmic shell of the channel shifts downward-outward when going from the closed to the open state. To facilitate visualization only the front protomer is shown in colors, while the other three protomers are shown as gray transparent volumes. The sarcoplasmic reticulum membranes are shown as black discs. **b** Overlapped models of closed RyR2 (PDB:7UA5, gray) and subprimed RyR2-S2808D (PDB:8UQ2, magenta). The arrows show that the cytoplasmic shell of the channel shifts downward-outward when going from the closed to the subprimed state. **c** RMSD normalized projection of WT RyR2 and RyR2-S2808D channels with different treatments. Values close to 0 or 1 indicate the conformation of the cytoplasmic shell is more similar to the closed WT RyR2 (PDB:7UA5), or the open WT RyR2 (PDB:7UA9), respectively. Negative values indicate the conformation of the cytoplasmic shell is more upward-inward than the closed state. **d** Overlapped models of subprimed RyR2-S2808D (PDB:8UQ2, magenta) and closed RyR2-S2808D + ARM210 (PDB:8UQ3, cyan). The arrows show that the cytoplasmic shell of the channel in the presence of ARM210 shifts upward-inward reversing the primed state back towards the closed state. **e** Same as (**b**) but with primed RyR2-S2808D + rapamycin (PDB:8UQ5, green). **f, g** cryo-EM maps focused on casltabin-2 (cyan) binding site of RyR2-S2808D (**f**) and RyR2-S2808D + rapamycin (**g**).

unable to detect any additional density in the cryo-EM map of RyR2-S2808D that could explain the incorporation of irreversible oxidative modifications due to the small size of the oxidative modifications (usually one oxygen atom), which is below the resolution limit achieved by cryo-EM.

Treatment with the Rycal drug ARM210 reversed these structural changes in RyR2-S2808D, resulting in a conformation of the cytoplasmic shell that is more upward-inward than the closed state of WT RyR2 (RMSD = -0.08, Fig. 1c, d). To induce reversible oxidative modifications, S-nitrosylation, and glutathionylation we added $H_2O_2$, NOC-12, and glutathione, respectively, to the purified RyR2 protein. However, the reaction time was limited to 30 minutes to avoid aggregation and denaturation of RyR2. Treatment with $H_2O_2$, NOC-12, and glutathione also rendered the cytoplasmic shell in a subprimed state, slightly more downward-outward than the control (RMSD = 0.13, Fig. 1c). As with RyR2-S2808D, we were unable to detect any additional cryo-EM density to attribute to post-translational modifications due to the limits of the resolution obtained from the cryo-EM experiments. This was because of the reduced number of resolvable cryo-EM particles (1.4 particles/ micrograph for the H/N/G condition vs an average of 12 particles/ micrograph for the other conditions) likely due to RyR2 aggregation induced by the oxidizing agents (Supplementary Fig. 2b). However, we hypothesize that in vivo the membrane-embedded channels would be more resistant to aggregation and denaturation and a more pronounced primed state could be achieved thus increasing the diastolic SR $Ca^{2+}$ leak.

Finally, we added rapamycin to compete the calstabin-2 off of RyR2 to simulate the calstabin-2 depletion of RyR2 observed in failing hearts[13]. Rapamycin is a macrocyclic drug that specifically binds to calstabins in the same site as RyRs do, sequestering calstabins and preventing their interaction with RyRs. Therefore, it prevents calstabin binding to RyRs. As expected, treatment with rapamycin rendered channels depleted of calstabin-2 (Supplementary Fig. 2b). We observed that RyR2-S2808D totally depleted of calstabin-2 is in a primed state comparable to the CPVT mutants, consistent with leaky arrhythmogenic RyR2 channels (RMSD = 0.30, Fig. 1c,e-g). We analyzed the structural differences to determine the molecular mechanism by which calstabin-2 stabilizes the upward-inward conformation of the cytoplasmic shell found in the closed state. From a structural point of view, calstabin-2 binds to a multidomain surface composed of the NSol, SPRY1, SPRY3, and JSol domains (Supplementary Fig. 5a). These domains rotate relative to each other between the closed (most upward-inward) and open (most downward-outward) states, suggesting that the decreased fit for calstabin-2 in the primed (the present study) and open[45] states is due to the relative rotation of the NSol, SPRY1, SPRY3, and JSol domains (Supplementary Fig. 5b, c). In the absence of calstabin-2, the changes in the calstabin binding site go in the same direction as those seen for the CPVT channels (Supplementary Fig. 5d). Therefore, we propose that calstabin-2 stabilizes the closed state of RyR2 by preventing the relative rotation of the NSol, SPRY1, SPRY3, and JSol domains, thus sterically blocking the most downward-outward conformations of the cytoplasmic shell found in the primed and open states. We also analyzed the cryo-EM structures

of RyR2 previously published by others[46,47], reaching the same conclusion (Supplementary Fig. 5e, f). In skeletal muscle, it has been reported that, similar to RyR2, the affinity of RyR1 for calstabin-1 (FKBP1A or FKBP12) is higher in the closed state compared to the open state[48] and that absence of calstabin-1 sets RyR1 in the primed state[49]. Therefore, we analyzed a disease-linked RyR1 mutant channel[50], which is also in a pathological primed state, and found the same changes in the calstabin binding site (Supplementary Fig. 5g), suggesting this is a conserved mechanism of calstabin binding and stabilization shared by the two RyR isoforms. Finally, it has been reported that calstabin-1, which is also expressed in the heart, binds and stabilizes RyR2 but with lower affinity than calstabin-2[51,52]. We analyzed the structures of the complexes between RyRs and calstabins and determined that the identical structure and mode of binding could explain the common stabilizing role that both calstabins have on RyR2, and that two negative residues (E32 and D33 in calstabin-1 vs Q32 and N33 in calstabin-2) could explain the lower affinity of RyR2 for calstabin-1 due to electrostatic repulsion (Supplementary Fig. 6).

## CPVT-associated RyR2-R420Q and RyR2-R420W are in the primed state

We chose the CPVT-related RyR2-R420Q and RyR2-R420W mutants because they are among the most frequent mutations reported in CPVT patients[53]. Functional studies of these mutations have shown afterdepolarizations, spontaneous sparks, and other indicators of increased diastolic SR $Ca^{2+}$ leak[34–41]. We sought to determine whether these CPVT-linked mutations result in the pathological primed state. Moreover, by analyzing two different mutations at the same residue, we were able to determine whether the identity of the mutant residue differentially affects the local and global structures. Recombinant human isoforms of the PKA-phosphorylated CPVT-linked RyR2-R420Q and RyR2-R420W channels were expressed in HEK-293 cells and purified as previously described (Supplementary Fig. 7)[10]. We determined the cryo-EM structures of the mutant channels under conditions that simulate diastole (e.g. normally non-activating nanomolar cytosolic $[Ca^{2+}]$, Supplementary Fig. 2c, d, 3b, Supplementary Table 2). Under these conditions, we found that the pore of the RyR2-R420Q and RyR2-R420W channels remained closed (Supplementary Fig. 4), but the cytoplasmic shell exhibited a downward-outward movement compared to the closed WT RyR2, reaching the primed state (Supplementary Fig. 8a, b). Calculation of RMSD normalized projections showed that the RyR2-R420Q had more pronounced conformational changes compared to the RyR2-R420W, but smaller than that of another CPVT mutant channel that we previously reported, the RyR2-R2474S (Fig. 2)[10]. Treatment with the Rycal drug ARM210 partially reversed these movements, reaching a conformation closer to that of the closed state of WT RyR2 (Fig. 2, Supplementary Fig. 8c, d). Treatment with ARM210 reduced the RMSD values by 0.18 ($n = 4$, SD = 0.025, $p = 0.0004$) for the three CPVT channels and the HF channel analyzed, strongly confirming a common inhibitory mechanism (Fig. 2). Similarly to RyR2-R2474S[10], we detected an increased density in the cleft of the RY1&2 domain of RyR2-R420Q, RyR2-R420W treated with ARM210, where Rycals were shown to bind (Supplementary Fig. 8e, f)[10,54]. We also detected an increased density in between the BSol1 and RY1&2 domains. This suggests that binding of ARM210 to the RY1&2 domain affects the local structure and stabilizes the interaction with the BSol1 domain, which in turn stabilizes the upward-inward conformation of the cytoplasmic shell associated with a stable closed pore of the channel in agreement with our previous reports (Supplementary Fig. 8e–h)[10,54].

We observed that R420Q and R420W mutations introduce a distinct mode of motion of the NTD-B domain. In WT RyR2, the relative orientation between the NTD-B and NSol domains remains unaffected between the closed and open states (Fig. 3a). However, the mutant channels in the primed state revealed that the NTD-B is rotated relative

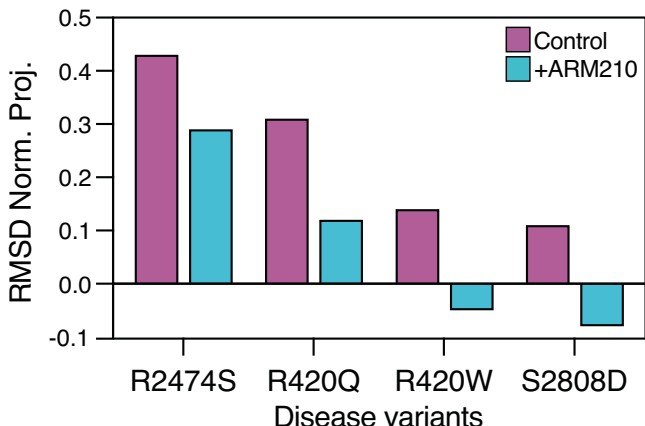

**Fig. 2 | Cryo-EM reconstructions of human RyR2 variants show that the primed state is reversed by treatment with the Rycal ARM210.** RMSD-normalized projections of the primed PKA-phosphorylated CPVT and HF variants in the absence and presence of ARM210. Values close to 0 or 1 indicate the conformation of the cytoplasmic shell is more similar to the closed WT RyR2, or the open WT RyR2, respectively. Negative values indicate the conformation of the cytoplasmic shell is more upward-inward than the closed state.

to the NSol (2-4°, Fig. 3b, c). The NTD-B domain stabilizes the closed state of WT RyR2 channel by interacting with the CSol and NTD-A⁻ domains but these stabilizing interactions are lost in the open state (Supplementary Fig. 9). In the mutant channels, the NTD-B domain rotates and separates itself from the CSol and the NTD-A⁻ domains, weakening the inter-domain interaction and promoting the primed state (Supplementary Fig. 9). In WT RyR2, R420, located in the NSol domain, interacts directly with the NTD-B domain (Supplementary Movie 1). According to the atomic model, the main interactions of R420 are a hydrogen bond with the backbone of V300 and a π-π stacking with R298, in agreement with previous crystallographic data (Fig. 3d)[55,56]. In RyR2-R420Q, mutation of R420 to the smaller residue Q420 results in the loss of both interactions made by R420 in WT RyR2. In this case, disruption of such interactions results in a 0.7 Å downward movement of the loop formed by the residues 298-304, which contains both interacting residues R298 and V300 (Fig. 3e), leading to the rotation of the whole NTD-B domain. In RyR2-R420W, the W420 residue also results in the loss of both interactions made by R420 in WT RyR2. However, due to its larger size and steric effect, the R420W mutation results in a 0.4 Å downward movement of the 298-304 loop (Fig. 3f), which might be not significant as it is close to the uncertainty level, measured as the RMSD between aligned NSol domains (0.34 Å), causing a less pronounced but significant rotation of the NTD-B compared to RyR2-R420Q (Fig. 3b,c).

## Channels in the primed state have reduced calstabin-2 affinity

Calstabin-2 depletion from the RyR2 macromolecular complex increases the open probability of RyR2 under conditions where the channel is supposed to be tightly closed, resulting in a pathological leak of intracellular $Ca^{2+}$. Calstabin-2 binding is reduced in CPVT mutant RyR2 channels and in RyR2 channels with stress-induced post-translational modifications associated with HF[45,57,58]. We assessed the binding of calstabin-2 to the closed state in WT RyR2, and the primed state in the CPVT variants RyR2-R420Q, RyR2-R420W, and RyR2-R2474S. As expected, we found that the primed state exhibited weakened affinity for calstabin-2 (50-100% increase in $K_d$ caused by increased $k_{off}$) and treatment with ARM210, which reverses the primed state, increased the affinity of calstabin-2 to RyR2 (Fig. 4a, b). The decreased calstabin-2 binding to RyR2 due to increased $k_{off}$ in the primed states channels, while $k_{on}$ remains unchanged, is in agreement

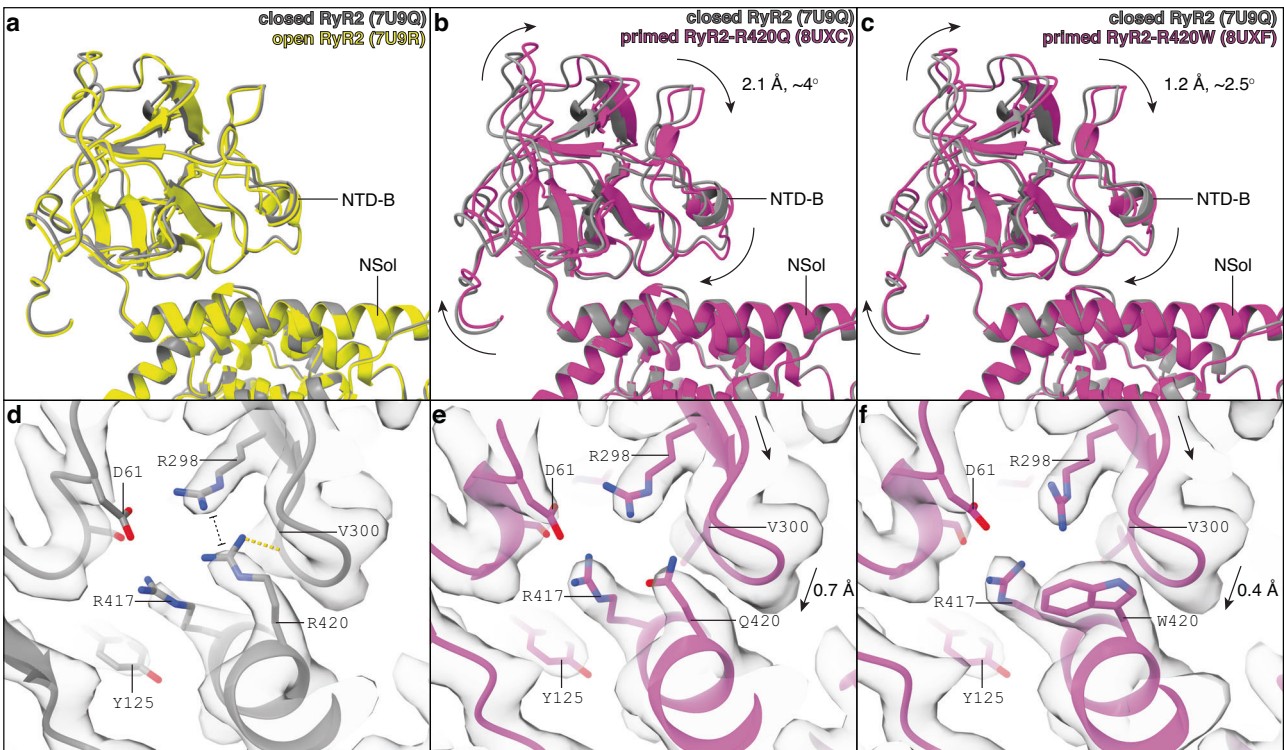

**Fig. 3 | R420Q/W mutations induce NTD-B domain rotation. a–c** Aligned models focused on the NTD-B domain of closed PKA-phosphorylated RyR2 (PDB:7U9Q, gray), open PKA-phosphorylated RyR2 (PDB:7U9R, yellow, **a**), primed PKA-phosphorylated RyR2-R420Q (PDB:8UXC, magenta, **b**), and primed PKA-phosphorylated RyR2-R420W (PDB:8UXF, magenta, **c**). Models were aligned at the NSol domain. Conformational changes are indicated with arrows. **d–f** The same models with overlapping cryo-EM maps around residue 420 of closed PKA-phosphorylated RyR2 (**d**), primed PKA-phosphorylated RyR2-R420Q (**e**), and primed PKA-phosphorylated RyR2-R420W (**f**). R420 interacts with R298 and V300, generating a network of stabilizing interactions. The presence of smaller residues in the mutants leads to the collapse of the 298-303 loop. Conformational changes of the 298-303 loop are indicated using arrows, and distances are provided.

with previous reports of different RyR2 states exhibiting changes in $k_{off}$ but not in $k_{on}$[45]. We further analyzed the effect of PKA phosphorylation of RyR2, which increased $k_{off}$ and $K_d$ for both WT and CPVT RyR2 variants (Fig. 4b). For PKA-phosphorylated WT RyR2 the increase in $k_{off}$ and $K_d$ was around 70% compared to the control, while for the PKA-phosphorylated CPVT RyR2 variants the increase was closer to ~100% suggesting a synergic effect between PKA phosphorylation and CPVT mutations, explaining the marked effects of intense exercise or emotional stress which causes RyR2 mediated SR Ca²⁺ leak in these patients. We also analyzed the effect of phosphomimetics and oxidation (Fig. 4b). Both RyR2-S2808D and WT RyR2 treated with $H_2O_2$ exhibited increased $k_{off}$ and $K_d$ of ~100%, but the combination was not additive. Interestingly, the values of $k_{off}$ and $K_d$ for RyR2-S2808D treated with $H_2O_2$ are similar to those of PKA phosphorylated CPVT variants, suggesting similar structural and arrhythmogenic properties, in agreement with our structural data.

Together, the cryo-EM structural data, and the $k_{off}$ and the $K_d$ data suggest that the primed state of the RyR2 variants favors the dissociation of calstabin-2. Moreover, there is a direct correlation between the RMSD value of the RyR2 variants and their calstabin-2 affinity (Fig. 4c). Interestingly, the RMSD values and calstabin-2 affinity exhibit a trend towards inverse proportionality compared to the average age of onset of symptoms reported in our CPVT database (Fig. 4c)[53]. This suggests that the magnitude of structural changes induced by the CPVT-linked mutants is associated with the level of pathogenicity of the disease. To further test this hypothesis, we analyzed the calstabin-2 affinity for other CPVT mutants with strong clinical information (Supplementary Fig. 10). Both the calculated $K_d$ and $k_{off}$ correlated with the age of onset of symptoms (Fig. 4d, e). This correlation suggests that calstabin binding affinity to RyR2, and

therefore the level of primed state, correlates with the pathogenicity of the disease and could be a marker for pathogenicity of novel RyR2 variants. In addition, these data further support a role for the primed state of RyR2 in HF and arrhythmias.

## Structural basis of abnormal systolic Ca²⁺ transients

Another common feature between HF and most CPVT cases are the abnormal systolic Ca²⁺ transients, usually characterized by a delayed peak with lower amplitude and a prolonged decay, suggesting uncoordinated or uncoupled channel gating[35,59–61]. Since RyR2-R420W presents abnormal systolic Ca²⁺ transients[35,59], we decided to analyze the effects of systolic concentrations of Ca²⁺ on the structure of RyR2-R420W. We solved the cryo-EM structure of the RyR2-R420W channels under conditions that resemble the Ca²⁺-induced Ca²⁺ release (CICR) that occurs during systole (Supplementary Figs. 2e, f, 3c). At the beginning of the systole, when CICR is the highest, the local Ca²⁺ concentration in the dyadic cleft reaches 10–100 μM or higher[62,63]. Here, we used 40 μM because this is the concentration at which ryanodine binding and lipid bilayer experiments result in maximum open probability, preventing RyR2 from reaching the closed or inactivated states. Under these conditions, RyR2-R420W + Ca²⁺ reached both primed (27% of the particles) and open (73% of the particles) states (Fig. 5a), suggesting increased open probability and Ca²⁺ sensitivity. This contrasts with what was observed for WT RyR2, where ATP and Ca²⁺ were unable to open the channel (100% of the particles in the primed state)[64]. The pore remained unaltered compared to the WT RyR2 states, confirming that the pore exists in two discreet conformations independent of the conformation of cytoplasmic shell or ligands bound (Supplementary Fig. 4i,j). The cytoplasmic shell of primed RyR2-R420W + Ca²⁺ moved downward-outward compared to

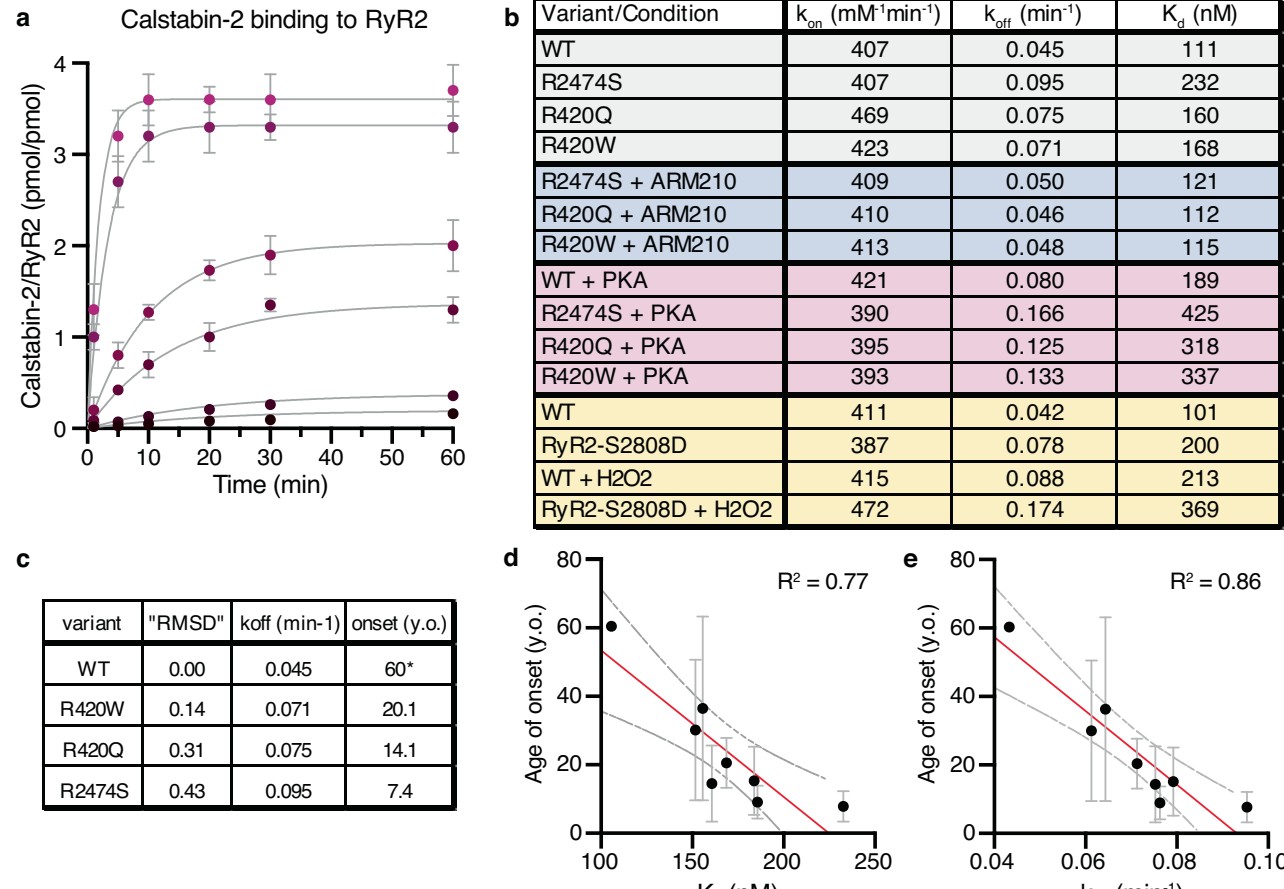

**Fig. 4 | Reduced calstabin-2 binding to the primed state of CPVT variants.**
**a** Binding curves of calstabin-2 to WT RyR2. Concentration of calstabin-2 is 5, 10, 50, 100, 500, and 1000 nM (from black to magenta), and RyR2 is 5 nM. Reactions were performed as duplicates. Data are presented as mean values and error bars as SD. All points were simultaneously fitted using the equation Association Kinetics - Two or more concentrations of ligand in GraphPad. Fitting curves are shown in gray, $R^2 = 0.9895$. **b** Table showing the values of $k_{on}$, $k_{off}$, and $K_d$ derived from the association kinetics curves for the different variants of RyR2. **c** Table showing the correlation between the primed state conformation ("RMSD"), calstabin-2 $k_{off}$, and onset age for the CPVT mutants tested (y.o.: years old)[53]. "RMSD" stands for the value of the normalized RMSD difference analysis of each structure. *The age of onset for WT RyR2 was considered as the average age when exercise-induced ventricular arrhythmias begin to be significant in the asymptomatic population[86,87]. **d, e** Correlation between $K_d$ and age of onset (**d**) and between $k_{off}$ and age of onset (**e**) for the variants tested. Reactions were performed as duplicates. Data are presented as mean values and error bars as SD.

the primed RyR2-R420W, suggesting that binding of $Ca^{2+}$ further increases the movement of the cytoplasmic shell as expected (Fig. 5b, d). The cytoplasmic shell of the open RyR2-R420W + $Ca^{2+}$ moves further downward-outward, compared to the primed state, reaching a conformation similar to that of the open WT RyR2 (Fig. 5d, Supplementary Fig. 11a). Surprisingly, the BSol2 domain was not resolved in either the primed or open states, suggesting that the increased dynamics of the cytoplasmic shell induced by both the CPVT mutation and $Ca^{2+}$ destabilizes the metastable BSol2 domain (Fig. 5e, f). The Bsol2 domain has been proposed to be the main contact point of RyRs self-association, and therefore to play a role in coupled-gating, and signal amplification[65,66]. Destabilization of the BSol2 domain during systole in disease-related RyR2s could explain loss of coupled gating resulting in delayed and reduced $Ca^{2+}$ transients peaks[67].

Addition of the RyR2 accessory protein CaM to RyR2-R420W + $Ca^{2+}$ had multiple stabilizing effects, first by reducing the open probability (61% of the particles in the primed state and 39% of the particles in the open state compared to 27% and 73% of RyR2-R420W + $Ca^{2+}$, respectively, Fig. 5a), second by moving the cytoplasmic shell upwards-inwards, both in the primed and open states (Fig. 5c, d, Supplementary Fig. 11b, c), and third by stabilizing the BSol2 domain, as seen by the cryo-EM maps (Fig. 5g). Compared to the previously

published structure of $Ca^{2+}$-CaM bound to RyR2, where the C-lobe of CaM was barely resolved[47], we observed a very well-defined density for the entire CaM protein allowing us to better model this unique mode of binding (Supplementary Movie 2). Our improved sample preparation and increased resolution allowed us to clearly detect the anchoring residues W3587, L3590, V3599 and F3603 in the CaM hydrophobic pockets (Supplementary Fig. 11d, Supplementary Movie 2), and to identify the inhibitory mechanism of $Ca^{2+}$-CaM binding to RyR2. Binding of $Ca^{2+}$-CaM not only stabilizes the CAMBD2 helix and the BSol2 domain, but also generates a conformational change in the JSol residues 1950–1960 attaching them to both the CAMBD2 and CaM N-lobe (Fig. 5i, Supplementary Fig. 11d–f). This conformational change of the residues 1950–1960 is stabilized mainly with weak hydrophobic interactions since most of the residues are Ala, Met, or Leu (Supplementary Fig. 11e, Supplementary Movie 2). This conformational change promotes the upward movement of the entire JSol domain, which in turns stabilizes the closed state (Fig. 5h, i). The upward movement is spread to the neighboring CSol domain, which induces a 0.8 Å expansion of the $Ca^{2+}$ binding site, measured as the distance between Cα of E3922 and T4931 (10.4 Å in primed RyR2-R420W + $Ca^{2+}$ and 11.2 Å in primed RyR2-R420W + $Ca^{2+}$ + CaM). This expansion of the $Ca^{2+}$ binding site most likely reverses the increased

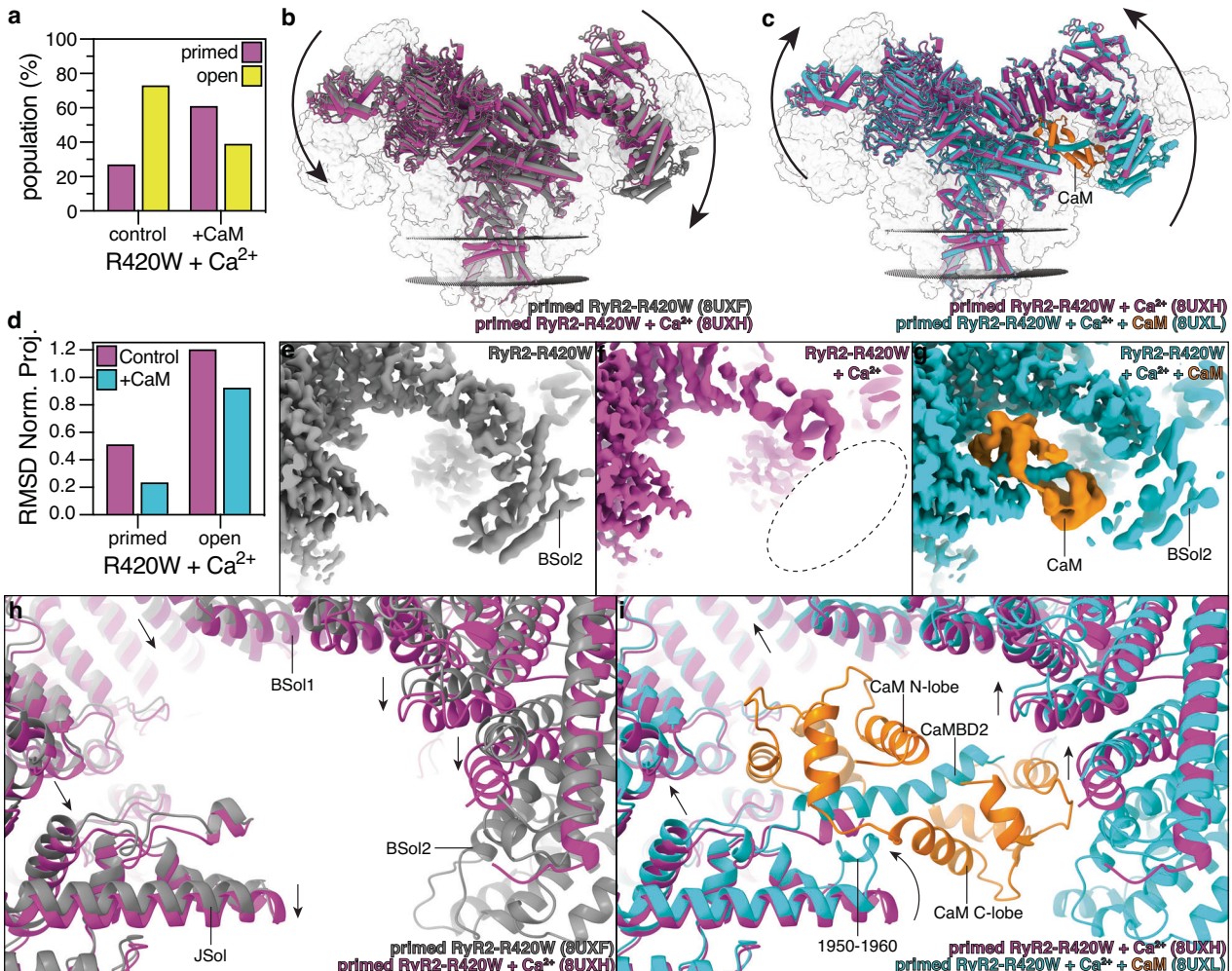

**Fig. 5 | RyR2-R420W increased sensitivity to Ca²⁺ is partially reversed by CaM binding. a** Cryo-EM particle population percentage in the samples of PKA-phosphorylated RyR2-R420W + Ca²⁺ in the absence and presence of CaM. **b** Overlapping models of primed PKA-phosphorylated RyR2-R420W (PDB:8UXF, gray) and primed PKA-phosphorylated RyR2-R420W + Ca²⁺ (PDB:8UXH, magenta). The arrows show that the cytoplasmic shell of the channel shifts downward-outward in the presence of Ca²⁺ inducing a more pronounced primed state. **c** Overlapping models of primed PKA-phosphorylated RyR2-R420W + Ca²⁺ (PDB:8UXH, magenta) and primed PKA-phosphorylated RyR2-R420W + Ca²⁺ + CaM (PDB:8UXL, cyan). The arrows show that the cytoplasmic shell of the channel shifts upward-inward in the presence of CaM, partially reversing the changes introduced by Ca²⁺. **d** RMSD normalized projection of RyR2-R420W + Ca²⁺ different states. Values close to 0 or 1 indicate the conformation of the cytoplasmic shell is more similar to the closed WT RyR2, or the open WT RyR2, respectively. **e**–**g** Cryo-EM maps of primed PKA-phosphorylated RyR2-R420W (**e**, gray), primed PKA-phosphorylated RyR2-R420W + Ca²⁺ (**f**, magenta), and primed PKA-phosphorylated RyR2-R420W + Ca²⁺ + CaM (**g**, cyan), highlighting the absence of the BSol2 domain (**f**, dashed circle) and the presence of CaM (**g**, orange). **h, i** Aligned models focused on the CaM binding site of primed PKA-phosphorylated RyR2-R420W (gray), primed PKA-phosphorylated RyR2-R420W + Ca²⁺ (magenta), and primed PKA-phosphorylated RyR2-R420W + Ca²⁺ + CaM (cyan). Models were aligned at the TM domain. Conformational changes are indicated with arrows.

sensitivity and affinity for Ca²⁺. Moreover, the presence of oxidation-sensitive Met residues in the binding interface might explain the reduced CaM affinity under oxidative stress, preventing stabilization of the BSol2 domain by Ca²⁺-CaM in disease conditions.

## Understanding the regulation of RyR gating by the cytoplasmic shell

We propose a model of how the binding of calstabin, calmodulin, and ligands, and the presence of mutations and post-translational modifications in the cytoplasmic shell allosterically regulate the pore and thus the open probability of RyRs. This model simplifies the understanding of the structure-function of RyRs proposing that it can be separated into two functional parts, the pore domain that opens to release Ca²⁺ from the ER/SR ("gating") and the cytoplasmic shell which regulates the gating of the pore domain by adopting more upward-inward conformations (prevents gating and leads to the closed state)

or more downward-outward conformations (promotes gating and leads to the open state). To describe the thermodynamics of the conformational space of the cytoplasmic shell, inspired by a previous work on RyR1 cryo-EM single particle manifold-based analysis[68], we propose two components: a flexion component and a pore state component (Fig. 6a–c). The flexion component of the cytoplasmic shell conformational space is an isoenergetic continuum of rapidly interchanging conformations, since only rotation of domains without rearrangement of bonds is involved and evidenced by the high heterogeneity observed in cryo-EM (Fig. 6b black). The pore state component of the cytoplasmic shell conformational space has two local minimums that correlate with the closed state, which has a more upward-inward conformation, and the open state, with more downward-outward conformation (Fig. 6c black). While the pore state is regulated by Ca²⁺ binding, which switches the global minimum from the closed state to the open state (Fig. 6c orange), the cytoplasmic

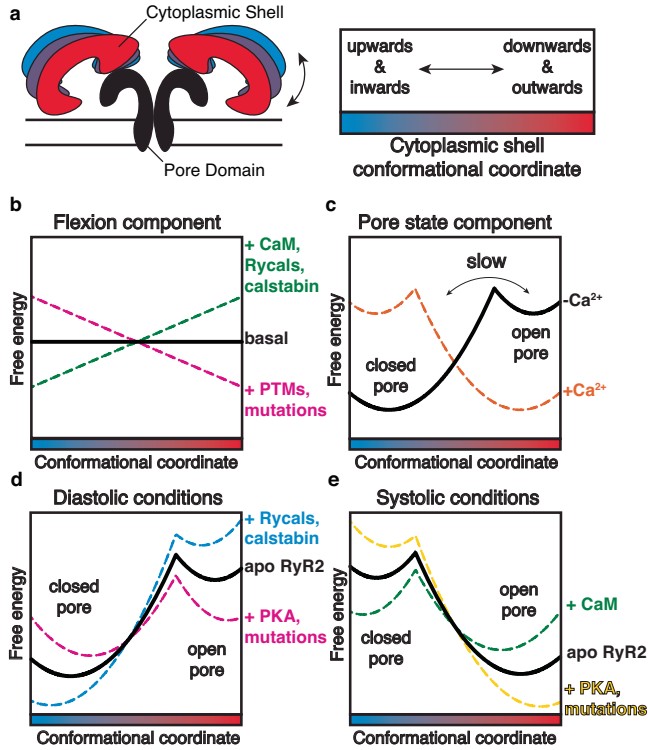

**Fig. 6 | Regulation of RyR gating by the cytoplasmic shell. a** Schematic representation of the conformational space of the cytoplasmic shell. **b, c** Schematic view of the dynamic energy landscape of the flexion component (**b**) and pore state component (**c**) of the conformational space of the cytoplasmic shell. In the flexion component, all conformations are energetically equal in the basal state (black). Binding of the ligands creates an alternative landscape that penalizes (green) or favors (magenta) the most downward-outward conformations. RyR2 can jump between two landscapes by ligand binding/unbinding or post-translational modifications (phosphorylation/dephosphorylation). In the pore state component, in the absence of activating Ca²⁺, the global minimum is the closed state with upward-inward conformations of the cytoplasmic shell. In the presence of activating Ca²⁺, the global minimum is the open state with downward-outward conformations of the cytoplasmic shell. **d, e** Schematic view of the dynamic energy landscape of the conformational space of the cytoplasmic shell (sum of both components), which explains the regulatory control of the cytoplasmic shell on the population distribution of RyR2 during diastole (**d**) and systole (**e**). Figure 6a adapted from Marco C. Miotto et al., Structural analyzes of human ryanodine receptor type 2 channels reveal the mechanisms for sudden cardiac death and treatment. Sci. Adv. 8,eabo1272(2022). https://doi.org/10.1126/sciadv.abo1272 under a CC BY license: https://creativecommons.org/licenses/by/4.0/[10].

shell component is the target of multiple regulatory events. The binding of calstabin-2 induces an upward-inward movement, suggesting the most upward-inward conformations are structurally and thermodynamically stabilized. This is confirmed by our structural analysis, where calstabin-2 affects the relative rotation of neighboring domains, and by the calculated affinities, where the closed WT channels with the most upward-inward conformations have the highest affinity for calstabin-2 (lowest $K_d$) and hence the lowest energy. In contrast, the primed channels with more downward-outward conformations have larger $K_d$ with intermediate energies. Both the closed and primed states have the pore in a closed conformation, making calstabin-2 effects independent of the pore state component, and hence, affecting only the flexion component. Moreover, the open channels with the most downward-outward have the highest $K_d$ and the highest energy (Fig. 6b green)[45]. The induction of an upward-inward motion while maintaining a closed pore is also observed in the presence of CaM, Ca²⁺-CaM, or ARM210, suggesting they have similar effects on the energetic profile of the flexion component (Fig. 6b

green)[10]. On the other hand, introduction of CPVT mutations or post-translational modification, like PKA phosphorylation, induces downward-outward motions while maintaining a closed pore, suggesting that they have an opposite effect on the energy profile of the flexion component (Fig. 6b magenta). The global energy landscape of the cytoplasmic shell conformational space is obtained by adding both the flexion and pore state components (Fig. 6d, e). This proposed energy landscape could explain the different cytoplasmic shell conformations and the relative populations between closed and open states obtained by cryo-EM in this paper (Supplementary Fig. S12).

## Discussion

To better understand the mechanistic underpinnings that result in leaky RyR2 channels that play a role in heart failure and arrhythmogenic disorders, we analyzed the structures of the calstabin-depleted phosphomimetic RyR2-S2808D channel and two CPVT-linked mutant channel structures, RyR2-R420Q and RyR2-R420W. Each of these structures show that the modified channels are in the primed state, defined as an intermediate conformation of the cytoplasmic shell between the closed and open WT RyR2 channels. Moreover, we determined that the conformation of the cytoplasmic shell can be affected by mutations, post-translational modifications, and ligand/protein binding, suggesting it plays a role in the regulation of the pore stability. The primed state is such that RyR2 in a failing heart would be readily and inappropriately activated by stress conditions, resulting in uncontrolled diastolic SR Ca²⁺ leak and cardiac arrhythmias, which are the cause of death of most patients with HF (Fig. 7a,b). Moreover, both the CPVT-linked RyR2-R2474S channel[10], which is also in a primed state, as well as the RyR2 channels depleted of calstabin-2, promote atrial fibrillation[3,26,69,70] suggesting that the primed state reached by these channels explains the occurrence of arrythmias in HF as well as in CPVT and atrial fibrillation. RyR2-R420Q and RyR2-R420W mutations also set the channels into the primed state, although the magnitude of structural changes of the cytoplasmic shell is less pronounced compared to the previously reported RyR2-R2474S[10]. We found that the differences in the level of the downward-outward movement of the cytoplasmic shell correlate with the age of onset of CPVT patients, further supporting the importance of the primed state in the development of these pathologies. Treatment with rycals has been shown to prevent arrythmias[10], and as shown here, to revert the primed state towards the closed state, suggesting the primed state of RyR2 is the common denominator in the arrhythmic events present in HF, CPVT, and atrial fibrillation.

A question that arises is to why patients and mouse models harboring RyR2 CPVT mutations, which have channels in the primed state, do not develop HF like RyR2-S2808D mice do (reduced ejection fraction and cardiac remodeling). We postulate that impaired contractility in failing hearts is due in part to the diastolic SR Ca²⁺ leak that reduces SR Ca²⁺ stores that in turn is caused by the chronic neurohormonal response characteristic of HF (e.g. chronic hyperadrenergic state that does not exist in CPVT). This neurohormonal response also affects other excitation-contraction coupling components and results in remodeling of the dyadic nanoscale architecture[71,72]. On the other hand, CPVT patients exhibit arrhythmias only during acute and intense adrenergic stimulation when PKA, besides phosphorylating RyR2, activates SERCA2a, which increases SR Ca²⁺ uptake and SR Ca²⁺ load. Increasing SR Ca²⁺ uptake, in conjunction with leaky RyR2, results in significant diastolic SR Ca²⁺ leak, delayed afterdepolarizations, and fatal arrhythmias. Under resting conditions, the CPVT RyR2-related Ca²⁺ leak is likely not significant as channels are not phosphorylated and the SR Ca²⁺ load is not increased. Moreover, CPVT patients and mice are heterozygous, as homozygosity for CPVT mutations is lethal, leading to a mixed population of WT and CPVT RyR2 channels in vivo. Indeed, the post-translational modifications of RyR2 observed in failing hearts have the potential to affect all RyR2 channels, in contrast to

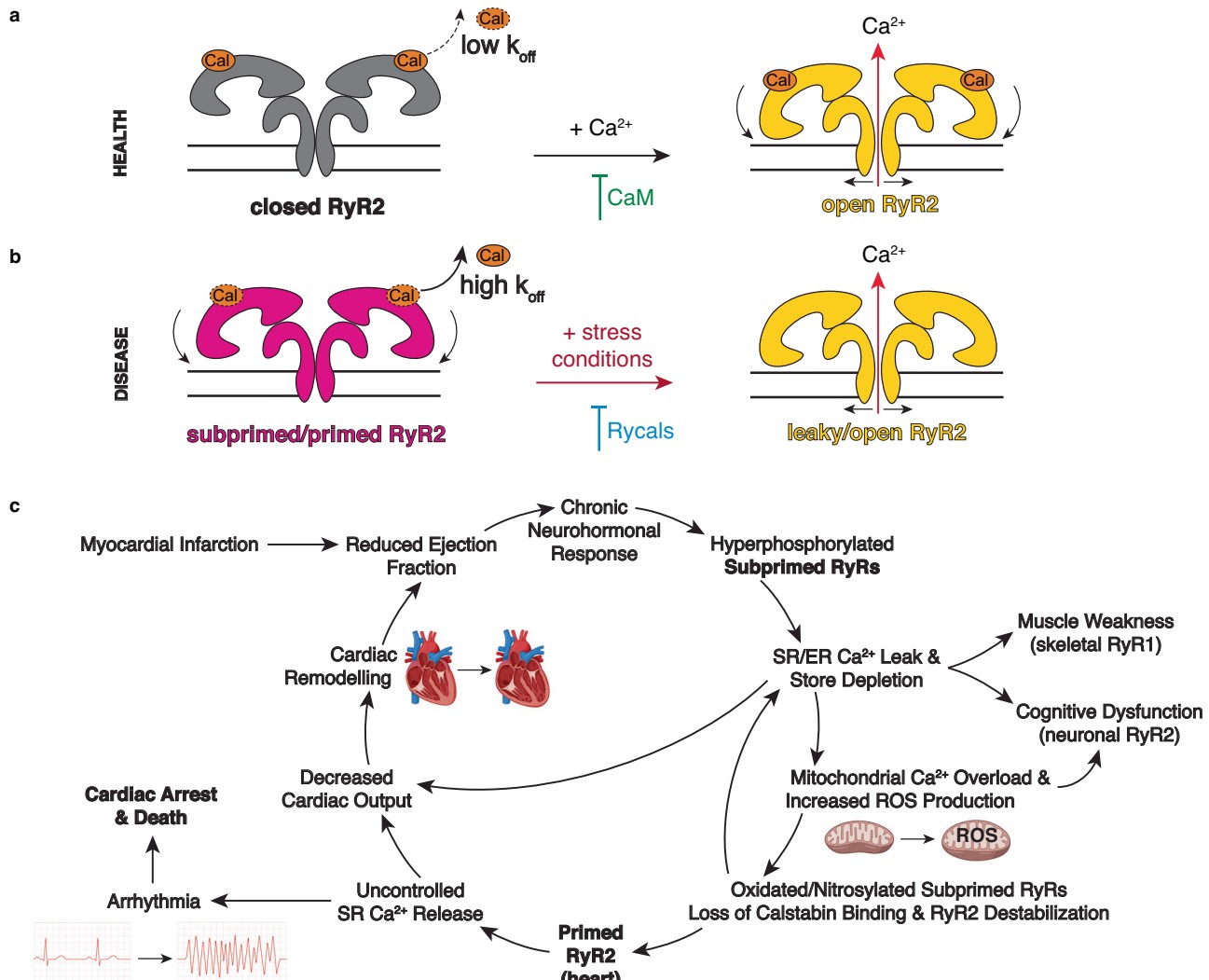

**Fig. 7 | Proposed structural mechanisms of leaky RyR2 and its involvement in disease. a, b** Schematic representation of the normal function of RyR2 (**a**) and the pathological disease-related disfunction (**b**). In the case of disease variants, the resting state is in a subprimed/primed state, which correlates with the higher calstabin-2 $k_{off}$ and $Ca^{2+}$ leak during diastole under stress conditions. The orange object labeled "Cal" stands for calstabin. Figure 7a,b adapted from Marco C. Miotto et al., Structural analyzes of human ryanodine receptor type 2 channels reveal the mechanisms for sudden cardiac death and treatment. Sci. Adv. 8,eabo1272(2022). https://doi.org/10.1126/sciadv.abo1272 under a CC BY license: https://creativecommons.org/licenses/by/4.0/.[10] **c** Vicious cycle involved in heart failure progression and arrhythmias. Figure 7c was created with BioRender.com released under a Creative Commons Attribution-NonCommercial-NoDerivs 4.0 International license https://creativecommons.org/licenses/by-nc-nd/4.0/deed.en.

heterozygous CPVT patients in whom not all channels may be leaky. Furthermore, we are analyzing the structure of CPVT mutant channels in their fully PKA-phosphorylated states. The data showing reduced calstabin-2 affinity indicate that PKA phosphorylation of CPVT channels has a synergistic effect in terms of decreasing calstabin-2 affinity and, consequently, the degree of the primed state. The calstabin-2 affinity for RyR2-S2808D was similar to that of dephosphorylated CPVT channels, consistent with a similar degree of the subprimed state. In the case of heterozygous patients or murine models, the number of channels in a subprimed state could be reduced due to mixing with WT channels, reaching states where the leak is insignificant under resting conditions, not substantial enough to cause SR $Ca^{2+}$ store depletion, dyadic nanoscale architectural remodeling, and reduced contractility. On the other hand, the RyR2-S2808D mutation induced leak would be worse in vivo due to RyR2 oxidation, nitrosylation, and calstabin-2 depletion, all of which occur in failing hearts, leading to increased primed state structures as observed by cryo-EM.

Finally, we found that PKA-phosphorylated WT RyR2 and oxidized RyR2-S2808D are in subprimed states. We reasoned that the physiological role of the subprimed state is to enable increased SR $Ca^{2+}$ release and enhanced contractility during the flight-or-fight response (PKA-phosphorylated RyR2). However, this same process becomes pathological during the chronic hyperadrenergic stimulation observed in HF. We propose that the subprimed state by itself is not sufficient to result in a large diastolic SR $Ca^{2+}$ leak and arrhythmias. However, hyperadrenergic-induced remodeled HF channels in the subprimed state promotes diastolic SR $Ca^{2+}$ leak that reduces the SR $Ca^{2+}$ store. This diminishes cardiac contractility and causes mitochondrial $Ca^{2+}$ overload, resulting in ROS production and sustained oxidative stress (Fig. 7c)[22]. Chronic oxidative stress could result in progressive sarcomere damage in both cardiac and skeletal muscles during heart failure[73]. This in combination with reduced SR $Ca^{2+}$ store, could result in skeletal muscle weakness, cognitive dysfunction, and decreased cardiac output, as seen in HF patients and promote a vicious cycle of

chronic neurohormonal response and SR Ca$^{2+}$ leak (Fig. 7c). Eventually, chronic diastolic SR Ca$^{2+}$ leak as occurs in HF (in contrast to episodic SR Ca$^{2+}$ leak as occurs in CPVT) would result in reduced Ca$^{2+}$ transients that cause cardiac dysfunction in failing hearts that is not seen in CPVT (Fig. 7c). Treatment with Rycals reduces RyR open probability and ER/SR Ca$^{2+}$ leak in diseased organs/tissues expressing the channel[73]. For instance, it has been shown by our group and others that Rycals are anti-arrhythmic, slow HF progression, improve skeletal muscle strength[14,73]. Interestingly, RyR2-S2808A mice harboring RyR2 channels that cannot be PKA phosphorylated are protected against HF progression following LAD ligation[15]. This further shows that the ability of RyR2 PKA phosphorylation to put the channel into the primed state combined with depleting the channel of calstabin-2 is a critical factor in promoting HF progression.

In the present study, we show that the beneficial effects of Rycals in HF are achieved by reversing the primed state and diminishing the RyR2-linked diastolic SR Ca$^{2+}$ leak, and not by preventing post-translational modifications, as previously shown[16,18,74–76]. The shared mechanism of RyR2-related SR Ca$^{2+}$ leak underscores the mechanistic and therapeutic connections between HF and arrhythmogenic disorders.

## Methods
### Generation and expression of RYR2-R420Q and RYR2-R420W mutants
The constructs expressing *RYR2-R420Q* and *RYR2-R420W* were formed by introducing the respective mutations into fragments of human *RYR2* using the QuikChange II XL Site-Directed Mutagenesis Kit (Agilent). Forward primers to introduce the mutations were, respectively: CAGCCCGAGTTATC(C**A**G)AGCACAGTCTTCCT and ACAGCCCGAGT-TATC(**T**GG)AGCACAGTCTTCC (mutated codon in parentheses, changed nucleotide in bold); the reverse primers for each construct were reverse complementary to these. Mutated fragments were subcloned into a full-length human *RYR2* construct in pcDNA3.1 vector using NotI and BstEII restriction enzymes, confirmed by sequencing. The construct expressing *RYR2-S2808D* was previously designed by our group[77]. For final expression, HEK293 cells grown in 150-mm dishes with Dulbecco's Modified Eagle Medium (DMEM) supplemented with 10% (v/v) fetal bovine serum (Invitrogen), penicillin (100 U/ml), streptomycin (100 μg/ml), and 2 mM L-glutamine were transfected with 25 mg per dish of human *RYR2-R420Q* or *RYR2-R420W* cDNA using PEI MAX (Polysciences) at a 1:5 ratio[78]. Cells were collected 48 hours after transfection.

### Purification of recombinant human RyR2 variants
All purification steps were performed on ice unless indicated. Approx. 40 dishes of HEK293 cells expressing human RyR2-R420Q, RyR2-R420W, or RyR2-S2808D were harvested by centrifugation for 10 min at 1500 g. The cell pellet was resuspended in tris malate buffer [10 mM tris malate (pH 6.8), 1 mM EGTA, 1 mM dithiothreitol (DTT), 1 mM benzamidine, 0.5 mM 4-benzenesulfonyl fluoride hydrochloride (AEBSF), and protease inhibitor cocktail] and was sonicated with six pulses of 30 s at 35% amplitude. The membrane fraction, containing RyR2, was centrifuged at 100,000 g for 30 min and the membrane pellet was resuspended with a glass homogenizer in CHAPS buffer [10 mM Hepes (pH 7.4), 1 M NaCl, 1.5% CHAPS, 0.5% phosphatidylcholine (PC), 1 mM EGTA, 2 mM DTT, 0.5 mM AEBSF, 1 mM benzamidine, and protease inhibitor cocktail]. The solubilized membrane solution was diluted 1:4 with the same buffer without NaCl to achieve a final concentration of 250 mM NaCl. To stabilize the RyR2 channels during the purification, 100 nmol of GST-calstabin-2 was added to both tris malate buffer and CHAPS buffer. The insoluble fraction was separated with a second centrifugation at 100,000 *g* for 30 min. The supernatant containing the CHAPS-solubilized RyR2 channels was filtered (0.22 μm filter) and loaded into a 5 ml HiTrap Q HP column

(Cytiva) previously equilibrated with buffer A [10 mM Hepes (pH 7.4), 0.4% CHAPS, 1 mM EGTA, 0.001% dioleoylphosphatidylcholine (DOPC), 230 mM NaCl, and 0.5 mM tris(2-carboxyethyl)phosphine (TCEP)]. The HiTrap Q HP column was eluted with a linear gradient between 300 and 600 mM NaCl. The fractions containing RyR2 mutant channels (300 to 350 mM NaCl) were pooled, and 100 nmol of GST-calstabin-2 was again added. The RyR2 fractions were loaded into a 1 ml GSTrap HP column (Cytiva), which was left recirculating overnight. The GSTrap HP column was washed with buffer A and eluted with glutathione buffer {10 mM Hepes (pH 8), 0.4% CHAPS, 1 mM EGTA, 0.001% DOPC, 200 mM NaCl, 10 mM GSH [glutathione (reduced form)], and 1 mM DTT. A second 1 ml HiTrap Q HP column (Cytiva), with the same conditions as before, was performed to separate the RyR2 mutant channels from the excess of unbound GST-calstabin-2 and GSH. Simultaneous cleavage of GST tag and PKA phosphorylation was performed by addition of 50 U of thrombin and 100 U of PKA (+ 10 mM EGTA, 8 mM MgCl$_2$, and 100 μM ATP for activity), respectively, for 30 min on ice. The sample was concentrated to 0.5 ml and a gel filtration step was run with TSKgel G4SW$_{XL}$ (TOSOH Biosciences) with buffer A. The RyR2 fractions were pooled and concentrated to a concentration of 2 to 4 mg/ml (with centrifugal filters of 100-kDa cutoff). To eliminate aggregates, the concentrated sample was filtered (with centrifugal filters of 0.22-μm cutoff). For the RyR2-R420Q and RyR2-R420W samples in "diastolic" conditions, 150 nM free Ca$^{2+}$ (600 μM total Ca$^{2+}$), 10 mM NaATP, and 500 μM ARM210 (when required) were added during the preparation of the final sample a few minutes before freezing the grids as published previously by our group[10,54]. For the RyR2-S2808D samples, 150 nM free Ca$^{2+}$ (600 μM total Ca$^{2+}$) and 10 mM NaATP were added to all samples. For the different conditions, 500 μM ARM210 (incubated with RyR2 for 30 min), a mix of 1 mM NOC-12, 1 mM GSH, and 0.3 mM H$_2$O$_2$ (preincubated for 2 h to release maximum NO species and incubated with RyR2 for 30 min), or 0.25 mM Rapamycin (incubated with RyR2 for 30 min) were added. For the RyR2-R420W samples in "systolic" conditions, 40 μM free Ca$^{2+}$ (1.7 mM total Ca$^{2+}$), 10 mM NaATP, and 20 μM CaM (when required) were added. MaxChelator webserver was used to calculate total/free Ca$^{2+}$ concentrations[79]. Quality control was assessed by SDS−polyacrylamide gel electrophoresis (SDS-PAGE). To assess total and phosphorylated RyR2, immunoblots were run using anti-RyR-34C (DSHB) and anti-RyR-pSer2808 (custom-made) antibodies, respectively[80].

### Purification of recombinant GST-calstabin-2
Recombinant human GST-calstabin-2 was expressed in BL21 (DE3) *Escherichia coli* cells with a thrombin protease cleavage site between GST and human calstabin-2 (FKBP1B). Protein expression was induced with 0.1 mM isopropyl-β-D-thiogalactopyranoside (IPTG) added to *E. coli* at an OD$_{600}$ (optical density at 600 nm) of 0.6–0.8 with overnight incubation at 18 °C before centrifugation at 6500 *g* for 10 min. The pellets were resuspended in buffer A (phosphate-buffered saline + 0.5 mM AEBSF) and lysed using an emulsiflex (Avestin EmulsiFlex-C3). The lysate was centrifuged for 10 min at 100,000 *g*. The supernatant was filtered (0.22 μm) and loaded into a 5 ml GSTrap HP column (Cytiva). The column was washed with 25 mL of buffer A before elution with buffer B [tris (pH 8), 2 mM DTT, 20 mM glutathione]. Fractions containing GST-calstabin-2 were pooled, concentrated, and dialyzed overnight in buffer A at 4 °C. The concentration of the final sample was determined using NanoDrop 1000 (Thermo Fisher Scientific) with absorbance at 280 nm and an extinction coefficient of 46,200 M$^{-1}$ cm$^{-1}$. The GST-calstabin-2 aliquots were stored at −80 °C.

### Purification of recombinant CaM
Human CaM was expressed in BL21 (DE3) *E. coli* cells with an N-terminal 6-histidine tag and a tobacco etch virus (TEV) protease

cleavage site. At an $OD_{600}$ of 0.6–0.8, protein expression was induced with 0.1 mM IPTG added to *E. coli* with overnight incubation at 18 °C before centrifugation for 10 min at 6500 g. The pellets were resuspended in buffer A [20 mM Hepes (pH 7.5), 150 mM NaCl, 20 mM imidazole, 5 mM 2-Mercaptoethanol, 0.5 mM AEBSF] and lysed using an emulsiflex (Avestin EmulsiFlex-C3). The lysate was centrifuged for 10 min at 100,000 g. The supernatant was loaded on a HisTrap HP column (Cytiva) and washed with 25 mL of buffer A before elution using a linear gradient from buffer A to buffer B (buffer A + 500 mM imidazole). The fractions containing CaM were pooled. 1 mg of TEV protease (Sigma-Aldrich) was added, and the mixture was dialyzed overnight at 4 °C into buffer C (buffer A with no imidazole). The sample was loaded on a HisTrap column with the flowthrough and the wash collected to retain fractions containing cleaved CaM before elution of TEV and any remaining contaminants with a linear gradient from buffer C to buffer B. The flowthrough and any fractions containing CaM were pooled, concentrated to >2 mM, which was determined by spectroscopy using NanoDrop 1000 (Thermo Fisher Scientific) with absorbance at 280 nm and the extinction coefficient of CaM (3000 $M^{-1}$ $cm^{-1}$). The CaM sample was stored at −80 °C.

## Cryo-EM sample preparation and data collection
The final sample (3 µL) was applied to UltrAuFoil holey gold grids (Quantifoil R 0.6/1.0, Au 300) previously cleaned with easiGlow (PELCO). Grids were blotted with ashless filter paper (Whatman) using blot force 10 and blot time 8 s before plunge-freezing into liquid ethane chilled with liquid nitrogen using Vitrobot Mark IV (Thermo Fisher Scientific) operated at 4 °C with 100% relative humidity.

High resolution data collection was performed at Columbia University on a Titan Krios 300-kV (Thermo Fisher Scientific) microscope equipped with an energy filter (slit width 20 eV) and a K3 direct electron detector (Gatan). Data were collected using Leginon[81] and at a nominal magnification of x105,000 in electron counting mode, corresponding to a pixel size of 0.83 Å. The electron dose rate was set to 16 e⁻/pixel per second with 2.5 s exposures for a total dose of 58 e/$Å^2$.

## Cryo-EM data processing and model building
Cryo-EM data processing was performed using cryoSPARC[82] with image stacks aligned using Patch motion, defocus value estimation by Patch CTF estimation. Particle picking was performed using the template picker with templates created from preexisting cryo-EM maps. Particles were subjected to 2D classification in cryoSPARC with 50–100 classes. Particles from the highest-resolution classes were pooled for ab initio 3D reconstruction with a single class followed by homogeneous refinement with C4 symmetry imposed. Heterogeneous refinement with two to six classes including "noise-like" volumes was performed to further separate particles from noise. A second round of heterogeneous refinement with four classes including "RyR-like" volumes was performed to further separate "good" particles from "bad" particles (partially misfolded particles that arise from the purification process or due to the interaction with the air-water interface). Since the RyR2 cytoplasmic shell is dynamic and shows a continuous range of conformations with a normal distribution around the average (Supplementary Fig. 13 and Supplementary Movies 3, 4), the classes of good particles, which have only small differences in the cytoplasmic shell upward-inward or downward-outward position, were pooled together in order to work with the global average and not discrete classes randomly separated by the software. Using a mask comprising the TM, a local heterogeneous refinement with six classes was performed to separate particles in the closed and open states. Non-uniform refinement was performed to estimate global resolution. C4 symmetry expansion was performed before local refinements. The masks used for local refinements were TaF + TM + CTD domains (residues 4131 to 4967), calstabin-2 + NTD + SPRY domains (residues 1

to 1646), JSol + CSol domains (residues 1700 to 2476,3590 to 4130), and BSol domain (residues 2400 to 3344). Only the TaF + TM + CTD mask used C4 symmetry. Smaller masks, for a second round of local refinement, were RY1&2 (residues 862 to 1076), RY3&4 (residues 2685 to 2909), BSol2 (residues 3042 to 3344), and CaM (when present in the sample). Local refinements used a dynamical mask with a far distance of 10 and 50 Å for the initial masks and small masks, respectively. The local resolution map of RyR2-S2808D, which is representative of all structures, is shown (Supplementary Fig. 14). Using the average volume from non-uniform refinement as base, the resulting local maps were combined in ChimeraX[83] to generate a composite map. We use the average map as base since RyRs have an intrinsic motion of the cytoplasmic shell with a normal distribution that is similar among variants and conditions (Supplementary Fig. 13). The calibration of the voxel size was performed using correlation coefficients with a map generated from the crystal structure of the NTD of RyR2 (4JKQ)[55]. Since the NTD contains the mutant site, only the NSol subdomain was used for the RyR2-R420Q and RyR2-R420W variants. The voxel size was altered by 0.001 Å per step, up to 20 steps in each direction.

Initial models were taken from the PDB and further modified: RyR2/calstabin-2 (PDB:7UA5) and CaM (PDB:6Y4O). Residue mutations, model fittings, and model building were performed in Coot[84], and final models were refined with Phenix tool RealSpaceRefine[85]. Cryo-EM statistics are summarized in Table S2. Figures of the structural analyzes were created using ChimeraX[83] and Adobe Illustrator 2021.

## RMSD normalized projection analysis
The analysis performed here aims to quantify the relative position of the cytoplasmic shell (using the residues 1-861 and 1077-2681 that correspond to the well resolved NTD, SPRY, JSol, and BSol1 domains, see local resolution on Supplementary Fig. 14) of an atomic model X from the closed (PDB: 7UA5) and open (PDB:7UA9) models of WT RyR2. To this end, the three models are aligned at the well resolved CSol domain (residues 3634-4016) and the RMSD of the Cα atoms are measured between each pair of atomic models: between the model X and the closed model RMSD(XC), between the model X and the open model RMSD(XO), and between the closed and open models RMSD(CO). Using a trigonometric approach, the RMSD normalized projection is calculated as the projection of the model X on the axis formed by the closed and open models, leading to the following equation (1):

$$RMSD\ norm.proj. = \frac{RMSD(CO)^2 + RMSD(XC)^2 - RMSD(XO)^2}{2RMSD(CO)^2}$$

Values of 0 or 1 indicate that the position of the cytoplasmic shell of the model X is identical to the closed state or open state of WT RyR2, respectively. Values between 0 and 1 indicate that the model X is in a primed state, somewhere between the closed and open states of WT RyR2. Values below 0 or above 1 indicate that the cytoplasmic shell of model X is more upward-inward than the closed state of WT RyR2 or more downward-outward than the open state of WT RyR2, respectively. Bar graphs were made with GraphPad Prism software. RMSD were measured in ChimeraX[83].

## Recombinant RyR2 microsomes generation
Microsomes from HEK293 cells expressing WT RyR2, or any of the mutant RyR2 were prepared by sonicating cell pellets in lysis buffer [10 mM tris (pH 7.0), 150 mM NaCl, 20 mM NaF, and protease inhibitors (Roche)]. The homogenate was centrifuged at 8000 g for 20 min at 4 °C, and the resulting supernatant was centrifuged at 37,000 g for 40 min at 4 °C. The final pellets, containing the microsomes, were resuspended in lysis buffer with addition of 250 mM sucrose, 10 mM Mops (pH 7.4), 10 mM EDTA. Samples were stored at −80 °C. To assess total and phosphorylated RyR2, immunoblots were run using anti-RyR-34C (DSHB) and anti-RyR-pSer2808 (custom made) antibodies,

respectively[80]. To assess total, nitrosylated, and oxidized RyR2, immunoblots were run using anti-RyR-5029 (custom made), Cys-NO ABM Y061263, and DNP Millipore Oxyblot (S7150) antibodies, respectively.

## Single-channel recordings

RyR2-R1500A-K1525A microsomes were fused to planar lipid bilayers formed by painting a lipid mixture of phosphatidylethanolamine and phosphatidylcholine (Avanti Polar Lipids) in a 5:3 ratio in decane across a 200 µm hole in polysulfonate cups (Warner Instruments) separating 2 chambers. The trans chamber (1.0 ml), representing the intra-SR (luminal) compartment, was connected to the head stage input of a bilayer voltage clamp amplifier. The cis chamber (1.0 ml), representing the cytoplasmic compartment, was held at virtual ground. Asymmetrical solutions used were as follows for the cis solution: 1 mM EGTA, 250/125 mM Hepes/Tris, 50 mM KCl, pH 7.35; and for the trans solution: 53 mM Ca(OH)$_2$, 50 mM KCl, 250 mM Hepes, pH 7.35. The concentration of free Ca$^{2+}$ in the cis chamber was calculated as previously described. Microsomes were added to the cis side and fusion with the lipid bilayer was induced by making the cis side hyperosmotic by the addition of 400–500 mM KCl. After the appearance of potassium and chloride channels, the cis side was perfused with the cis solution. At the end of each experiment, 10 µM ryanodine was added to block and confirm the identity of the RyR2 channels. Single-channel currents were recorded at 0 mV using a Bilayer Clamp BC-525D (Warner Instruments), filtered at 1 kHz using a Low-Pass Bessel Filter 8 Pole (Warner Instruments), and digitized at 4 kHz. All experiments were performed at room temperature (20 °C). Data acquisition was performed by using Digidata 1322 A and Axoscope 10.1 software (Axon Instruments). The recordings were analyzed using Clampfit 10.1 (Molecular Devices) and Graphpad Prism software.

## Calstabin-2 binding assay

Duplicate reactions containing 100 µg of recombinant RyR2 microsomes in 90 µl binding buffer (10 mM tris, 150 mM NaCl, pH 7.0) were preincubated for 30 min at room temperature with either buffer or 10 µM ARM210. The calstabin-2 binding reaction was initiated by the addition of 10 µl of various concentrations (5, 10, 50, 100, 500, 1000 nM) of $^{35}$S-labeled calstabin-2. Samples were incubated at room temperature and the reactions were stopped at various times (1, 5, 10, 20, 30, 60 min) by the addition of 1.0 ml ice-cold binding buffer followed by filtration through GF/B Whatman filters pre-equilibrated with 0.015% polyethylenimine. Filters were washed 3 times with 5 ml of wash buffer (10 mM MOPS, 200 mM NaCl, pH 7.4), dried, and counted in a scintillation counter. Nonspecific binding was determined using 10 µM Rapamycin. For each RyR2 variant, all data points were simultaneously fitted using the equation "Association Kinetics - Two or more concentrations of ligand" in GraphPad. Graphs were made using GraphPad.

## Reporting summary

Further information on research design is available in the Nature Portfolio Reporting Summary linked to this article.

## Data availability

Atomic models and cryo-EM maps are publicly available at wwPDB and EMDB under the following accession codes, respectively: 8UQ2 and EMD-42458 (RyR2-S2808D in the subprimed state), 8UQ3 and EMD-42459 (RyR2-S2808D + ARM210 in the closed state), 8UQ4 and EMD-42460 (RyR2-S2808D + H$_2$O$_2$/NOC-12/GSH in the subprimed state), 8UQ5 and EMD-42461 (RyR2-S2808D + rapamycin in the primed state), 8UXC and EMD-42759 (PKA-phosphorylated RyR2-R420Q in the primed state), 8UXE and EMD-42761 (PKA-phosphorylated RyR2-R420Q + ARM210 in the closed state), 8UXF and EMD-42762 (PKA-phosphorylated RyR2-R420W in the primed state), 8UXG and EMD-42763 (PKA-phosphorylated RyR2-R420W + ARM210 in the closed state), 8UXH and EMD-42764 (PKA-phosphorylated RyR2-R420W + Ca$^{2+}$ in the primed state), 8UXI and EMD-42765 (PKA-phosphorylated RyR2-R420W + Ca$^{2+}$ in the open state), 8UXL and EMD-42768 (PKA-phosphorylated RyR2-R420W + Ca$^{2+}$ + CaM in the primed state), 8UXM and EMD-42769 (PKA-phosphorylated RyR2-R420W + Ca$^{2+}$ + CaM in the open state). Source data are provided with this paper.

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

## Acknowledgements

Some of the studies reported herein were performed at the Columbia University Cryo-Electron Microscopy Center. We would like to acknowledge the molecular graphics and analyzes performed with UCSF ChimeraX, developed by the Resource for Biocomputing, Visualization, and Informatics at the University of California, San Francisco, with support from the National Institutes of Health R01-GM129325 and the Office of Cyber Infrastructure and Computational Biology, National Institute of Allergy and Infectious Diseases. We would like to acknowledge O. Clarke, F. Mancia, and members of their laboratories for helpful discussions and input, and R. Grassucci and Z. Zhang from the cryo-EM facility for their essential services. These studies were supported by NIH grants 5R01HL142903, 5R01DK118240, 5R01HL145473, 1RF1NS114570, R01NS124854, 1P01HL164319, and 5R01HL140934.

## Author contributions

Conceptualization: M.C.M. and A.R.M. Methodology: M.C.M., S.R., A.W., and A.R.M. Investigations: M.C.M., S.R., A.W., Q.Y., H.D., Y.L., G.W., and C.T. Validation: M.C.M., S.R., Q.Y., and G.W. Formal analyzes: M.C.M., S.R., and Q.Y. Resources: A.R.M. Data curation: M.C.M., S.R., and G.W. Writing—original draft: M.C.M. Writing—review and editing: M.C.M., A.W., H.D., and A.R.M. Visualization: M.C.M. Supervision: A.R.M. Project administration: A.R.M. Funding acquisition: A.R.M.

## Competing interests

A.R.M. is a member of the scientific advisory board and board of directors and an equity owner in ARMGO Pharma Inc. a biotech company developing RyR targeted therapeutics. Columbia University also owns equity in ARMGO Inc. A.R.M. has patent applications or pending or awarded patents, including U.S. 2014/0378437 and U.S. 7,718,644. The remaining authors declare no competing interests.
