## [Peer Review File · Nature Communications]

Structural basis for ryanodine receptor type 2 leak in heart failure and arrhythmogenic disordersREVIEWER COMMENTS

Reviewer #1 (Remarks to the Author):

The MS consists of a large array of CryoEM data and Calstabin binding data that, according to the authors, supports a unifying structural hypothesis for RyR2 dysregulation caused by multiple independent disease mechanisms. I have the following concerns and suggestions for the authors:

1. The MS is phenomenological. Somehow multiple interventions (oxidation, mutations, removal of subunit binding) induce somewhat similar overall conformational changes in human RyR2. With the exception of the two CPVT variants, exact molecular mechanism are not examined.
2. There is not enough detail provided in the methods for the work to be reproduced. For example, the authors do not provide information on dosage and duration of drug treatment with the Rycal. In its present form, the reader cannot evaluate the soundness of the experimental data.
3. Human RyR2 variants cause CPVT but not heart failure. This has been widely documented in the literature. As such, it is difficult to understand how there can be a unifying structural hypothesis that can be explained by reduced Calstabin binding, as the authors suggest.
4. Correlation data between calstabin binding affinity and severity of human CPVT: The authors claim a correlation without providing the clinical data and statistical analysis supporting the conclusions. To support the clinical data, a reprint is cited. Clinical severity of CPVT mutations is difficult to ascertain at best, and the data supporting the clinical onset of CPVT for the different variants are not included. These data would be better reported in a stand-alone MS.
5. Based on previous reports by other groups, native RyR2 in cardiac muscle have only a 10% occupancy of calstabin. As such, both the CryoEM and binding experiments using 100fold excess of calstabin to RyR2 are not representative of native RyR2.
6. How does Rycal binding to RyR2 stabilize the RyR2 structure? No explanation is provided. A detailed structure-activity analysis would be needed to support the claim that Rycal binding stabilizes the closed state in the various CPVT mutant channels.
7. The authors simulated the HF condition by incubating HEK cells in H₂O₂, resulting in reversible and irreversible oxidative modifications of RyR2. What molecular changes are induced is not examined. As such, there are multiple unknown modifications of the RyR2 channel that contribute to the altered CryoEM structure. Whether those modifications are prevented by Rycal treatment is not examined.
8. The structural data with the two CPVT variants and their effect on CaM binding are well-documented and could be reported as a stand-alone MS.

Reviewer #2 (Remarks to the Author):

The RYR2 mutants, S2808D, R420Q and R420W, are closely associated with cardiomyopathy and CPVT, two different types of heart disease. The authors here reported cryo-EM structures of these RYR2 mutants at high resolution. Through comparing the structures of these RYR2 mutants with that of RYR WT, the authors observed some structural differences in the cytoplasmic domains of RYR2. The authors further claimed that the mutations shift the RYR2 to the primed state that is easier to be activated. This explains how the mutations cause the unwanted leaking of the channel. Collectively, this work provides some molecular clues for how these mutations in RYR2 can cause heart disease. Overall, this is interesting work. The cryo-EM maps are of high quality. However, there are some concerns that need to be addressed. Here are my specific points:

(1) The authors compared the structure of RYR2 S2808D with previously determined RYR2 WT in the closed state. The RYR2 S2808D was purified from the cells under H₂O₂ treatment. If I understand correctly, RYR2 WT was purified from the cells without any H₂O₂ treatment. If this is the case, the structural comparison shown in Figure 1A and 1B would not be rigorous. We don't know for certain whether the detected conformational difference is caused by mutation or H₂O₂ treatment.

Similarly, the author mentioned that the cryo-EM structures of RYR2 R420Q/W were determined under conditions that simulate diastole. If the WT structure was not determined at the same condition, the comparison is not so meaningful.

(2) Based on Figure 1 and Figure 5, the peripheral domains undergo the largest movement between the WT and mutant RYR2. Nevertheless, the peripheral domains in the cryo-EM map of RYR2 mutants were resolved at relatively low resolution, indicating that these regions are very dynamic. It is not so meaningful to compare the conformation of a flexible domain between WT and mutant RYR2, because the resolved conformation is the average of many different conformational states. The authors need to perform 3D Variability Analysis with cryoSPARC to convince the readers that the observed conformational differences are not simply due to the intrinsic flexibility of the peripheral domains.

The authors need to show the local resolution map in the supplementary figures.

(3) As shown in Figure. 3, the authors observed certain movement of NTD-B domain in relative to NSol domain, between WT and mutant RYR. However, the observed conformational change might not be significant, as the movement is too small (0.7 and 0.4 angstrom). It could be simply result from the uncertainty in model building due to the limited resolution (~3 angstrom). The author mentioned that this conformational change is supported by the previous MD simulation result, but this claim lacks support by any figures. This part needs to be elaborated. Ideally, the authors need to perform the MD simulation by themselves to validate the conformational difference observed in the cryo-EM structures.

(4) During the image processing of each dataset, the authors merge all the classes that represent the closed state together. It is unclear to me why these classes can be combined. Do they have exactly the same conformation? Do the peripheral regions also adopt the same conformation? If not, the authors need to analyze each class individually.

Reviewer #3 (Remarks to the Author):

In this manuscript cryo-EM was used to compare the structures of cardiac isoform of the ryanodine receptor (RyR2) with various variants of RyR2 linked to heart failure. The authors found that mutations in RyR2 associated with heart failure make the channel more susceptible to a leak of calcium out of the sarcoplasmic reticulum. The leak occurs because the protein is placed in a primed state, between closed and open states, and binding of calstabin-2, a protein that stabilizes the closed RyR2, is reduced. This leak leads to afterdepolarizations, ventricular tachycardia, and sudden cardiac death. The authors also found that treatment with a Rycal drug helps maintain the channel in the closed state, which decreases the calcium leak and increases calstabin-2 binding.

Overall, this manuscript provides a very detailed analysis of RyR2 structures. As stated on line 411 these structures are consistent with previously published structures and similar conclusions were reached. The comparison between wild type RyR2 and several versions with pathogenic mutations provides ample justification for the selection of the mutations studied. The basic findings, that the mutations make RyR2 more leaky and that a Rycal drug can compensate for the altered structural changes, have been presented in previous work by this group. The specific mutations associated with known cardiac diseases makes the findings of particular interest to cardiologists, especially those developing treatments.

Line 161: explanation of the effects of rapamycin is needed.

Line 305: the observation that “the magnitude of structural changes induced by the CPVT-linked mutants is associated with the level of pathogenicity of the disease” provides strong support for the suggestions made by the authors. This finding should be further highlighted.

Line 333: the systolic calcium concentration is stated as 40 uM, but this seems too high if mimicking cytoplasmic calcium levels and too low for intra-SR calcium levels. More explanation for the selection of 40 uM should be provided.

Line 492: The authors assert that the “channel population with mixed WT:CPVT protomers (possible combination of protomers WT:CPVT would be 4:0, 3:1, 2:2, 1:3, and 0:4), with only 20% of the

channels having all four mutant protomers.” Unless the distribution was measured, the authors cannot assert that 20% of the channels would have only mutant protomers. The likelihood of preferential classes is high.

The object labeled “C” in Fig 7 and the intro figure needs to be defined in the legend.

Throughout the manuscript the authors use the phrase “suggesting it might” (or something similar). This phrase weakens the comment. Either the idea “suggests” the concept or “it might” influence the concept.

REVIEWER COMMENTS & AUTHOR RESPONSES

Reviewer #1 (Remarks to the Author):

The MS consists of a large array of CryoEM data and Calstabin binding data that, according to the authors, supports a unifying structural hypothesis for RyR2 dysregulation caused by multiple independent disease mechanisms. I have the following concerns and suggestions for the authors:

1. The MS is phenomenological. Somehow multiple interventions (oxidation, mutations, removal of subunit binding) induce somewhat similar overall conformational changes in human RyR2. With the exception of the two CPVT variants, exact molecular mechanism are not examined.

We noted in line 164 that we were unable to detect any significant modifications in the cryo-EM maps that allowed the determination of a clear molecular mechanism for RyR2-S2808D treated with oxidizing and nitrosylation agents. We now have revised the sentence to state: “As with RyR2-S2808D, we were unable to detect any additional cryo-EM density to attribute to post-translational modifications due to the limits of the resolution obtained from the cryo-EM experiments. This was because of the reduced number of resolvable cryo-EM particles (1.4 particles/micrograph for the H/N/G condition vs an average of 12 particles/micrograph for the other conditions) likely due to RyR2 aggregation induced by the oxidizing agents (Supplementary Fig. 2b).”

We also added a new sentence clarifying this for the RyR2-S2808D variant. The new sentence reads: “We were unable to detect any additional density in the cryo-EM map of RyR2-S2808D that could explain the incorporation of irreversible oxidative modifications due to the small size of the oxidative modifications (usually one oxygen atom), which is below the resolution limit achieved by cryo-EM.”

The molecular mechanism of the primed state by the removal of the subunit calstabin-2 was analyzed on lines 403-411 reaching to the conclusion that: “calstabin-2 stabilizes the closed state of RyR2 by sterically blocking the most outward and downward conformations of the cytoplasmic shell found in the primed and open states”. We realized that the analysis was in a different section of the results presentation, so we moved it to immediately follow the presentation of the RyR2-S2808D + Rapamycin variant cryo-EM results.

As mentioned by the reviewer, the molecular mechanism for the other 8 structures, that is the two CPVT variants in diastolic and systolic (absence and presence of Calmodulin) conditions, had been elucidated.

2. There is not enough detail provided in the methods for the work to be reproduced. For example, the authors do not provide information on dosage and duration of drug treatment with the Rycal. In its present form, the reader cannot evaluate the soundness of the experimental data.

We appreciate the reviewer’s thorough examination of the methods. Regarding the Rycal ARM210 conditions, these were already included in line 588 “500 μ M ARM210 (when required) was added”. We modified the sentence to make it clear that the drug was added during the last step

before freezing the grids consistently with our previous publications “For the RyR2-R420Q and RyR2-R420W samples in “diastolic” conditions, 150 nM free Ca²⁺ (600 μM total Ca²⁺), 10 mM NaATP, and 500 μM ARM210 (when required) were added during the preparation of the final sample a few minutes before freezing the grids as published previously by our group.^{10,54}”

For RyR2-S2808D samples, the dose was already included in line 589, but we now included the time of incubation “For the RyR2-S2808D samples, 150 nM free Ca²⁺ (600 μM total Ca²⁺) and 10 mM NaATP were added to all samples. For the different conditions, 500 μM ARM210 (incubated with RyR2 for 30 min), a mix of 1 mM NOC-12, 1 mM GSH, and 0.3 mM H₂O₂ (preincubated for 2 h to release maximum NO species and incubated with RyR2 for 30 min), or 0.25 mM Rapamycin (incubated with RyR2 for 30 min) were added.”

For the calstabin affinity experiments, details have been added to the methods. We now have included: “Duplicate reactions containing 100 μg of recombinant RyR2 microsomes in 90 μl binding buffer (10 mM tris, 150 mM NaCl, pH 7.0) were preincubated for 30 min at room temperature with either buffer or 10 μM ARM210.”

3. Human RyR2 variants cause CPVT but not heart failure. This has been widely documented in the literature. As such, it is difficult to understand how there can be a unifying structural hypothesis that can be explained by reduced Calstabin binding, as the authors suggest.

We think the reviewer misinterpreted our discussion, where we propose that the unifying structural mechanism is the primed state, similar between CPVT and calstabin-depleted RyR2, which is the cause of uncontrolled Ca²⁺ leak that leads to the arrhythmias present both in CPVT and HF. Reduced calstabin binding is not the unifying hypothesis but a consequence of the primed state and a valuable tool to analyze the degree of primed state of RyR2 variants. We have revised the abstract and discussion to make sure this is clear to the readers:

Abstract

“We propose a structural-physiological mechanism whereby the ryanodine receptor 2 channel primed state underlies the arrhythmias in heart failure and arrhythmogenic disorders.”

Discussion

“The primed state is such that RyR2 in a failing heart would be readily and inappropriately activated by stress conditions, resulting in uncontrolled diastolic SR Ca²⁺ leak and cardiac arrhythmias, which are the cause of death of most patients with HF (Figure 7a,b). Moreover, both the CPVT-linked RyR2-R2474S channel,¹⁰ which is also in a primed state, as well as the RyR2 channels depleted of calstabin-2, promote atrial fibrillation^{3,26,71,72} suggesting that the primed state reached by these channels explains the occurrence of arrhythmias in HF as well as in CPVT and atrial fibrillation... Treatment with rycals has been shown to prevent arrhythmias,¹⁰ and as shown here, to revert the primed state towards the closed state, suggesting the primed state of RyR2 is the common denominator in the arrhythmic events present in HF, CPVT, and atrial fibrillation.”

We are also aware that naturally occurring RyR2 variants cause CPVT but not HF, and we have had already addressed this in the original discussion in line 485. We have now expanded this section showcasing additional factors that differentiate CPVT and HF.

“A question that arises is to why patients and mouse models harboring RyR2 CPVT mutations, which have channels in the primed state, do not develop HF like RyR2-S2808D mice do (reduced ejection fraction and cardiac remodeling). We postulate that impaired contractility in failing hearts is due in part to the diastolic SR Ca^{2+} leak that reduces SR Ca^{2+} stores that in turn is caused by the chronic neurohormonal response characteristic of HF (e.g. chronic hyperadrenergic state that does not exist in CPVT). This neurohormonal response also affects other excitation-contraction coupling components and results in remodeling of the dyadic nanoscale architecture.^{73,74} On the other hand, CPVT patients exhibit arrhythmias only during acute and intense adrenergic stimulation when PKA, besides phosphorylating RyR2, activates SERCA2a, which increases SR Ca^{2+} uptake and SR Ca^{2+} load. Increasing SR Ca^{2+} uptake, in conjunction with leaky RyR2, results in significant diastolic SR Ca^{2+} leak, delayed afterdepolarizations, and fatal arrhythmias. Under resting conditions, the CPVT RyR2-related Ca^{2+} leak is likely not significant as channels are not phosphorylated and the SR Ca^{2+} load is not increased.”

4. Correlation data between calstabin binding affinity and severity of human CPVT: The authors claim a correlation without providing the clinical data and statistical analysis supporting the conclusions. To support the clinical data, a preprint is cited. Clinical severity of CPVT mutations is difficult to ascertain at best, and the data supporting the clinical onset of CPVT for the different variants are not included. These data would be better reported in a stand-alone MS.

We agree with the reviewer that the clinical severity of CPVT variants is difficult to assess and that is why we chose one of the clearest parameters which is onset age of symptoms. In our opinion, an onset at 5-years of age, for example, is more pathogenic and clinically relevant than an onset age at 25-years of age. The complete CPVT database upon which our study is based is now available on medRxiv: doi.org/10.1101/2024.03.15.24304349
<https://www.medrxiv.org/content/10.1101/2024.03.15.24304349v1.supplementary-material>

5. Based on previous reports by other groups, native RyR2 in cardiac muscle have only a 10% occupancy of calstabin. As such, both the CryoEM and binding experiments using 100fold excess of calstabin to RyR2 are not representative of native RyR2.

In our cryo-EM experiments, the molar ratio of calstabin-2 to RyR2 protomer is 1:1 (or 4 castabin-2 per RyR2 homotetramer). A molar excess of calstabin is used during the first steps of the purification of RyR2 but the excess calstabin is washed away during the last Q column (see Q₂ FT_{GSH} and Q₂ eluate in Supplementary Fig 1g), leaving only calstabin-2 that is bound to RyR2 (see the weak calstabin band compared to the RyR2 band in the final sample in Supplementary Fig 1g). For the binding experiments, a range of excess concentrations of calstabin is used to calculate kinetic and equilibrium constants. Such binding conditions are used commonly to determine binding affinities.

This reviewer’s comment suggests that calstabin binding to RyR2 has a minimal role on the normal physiology of the heart, although the role of calstabins has been strongly proven by multiple groups using a variety of approaches: 1) knockout mice;¹ 2) treatment with FK-506, a drug that specifically binds and sequesters calstabins which leads to increased Ca^{2+} transients and increased frequency of Ca^{2+} sparks.² Using indirect experiments, the reports mentioned by the reviewer

indicate that murine cardiac RyR2 would have 10-20% occupancy of calstabin-2, while the remaining 80-90% would be occupied by calstabin-1.^{3,4} However, there are other reports that show higher amounts of calstabin-2 in human cardiomyocytes and differences among mammalian species⁵⁻⁷ – the reported differences likely have to do with the differences in experimental approaches (e.g. basing relative protein amounts on studies using western blots which are notoriously difficult to quantify given differences in antibody avidity) vs radiolabeled binding studies which are likely to be more quantitative. Recent reports indicate that both calstabin-1 (FKBP12) and calstabin-2 (FKBP12.6) at saturating micromolar concentrations have similar inhibitory and stabilizing effects, suggesting that it would be indifferent which calstabin isoform is bound as long as the calstabin sites are occupied.^{8,9}

Response References:

- 1 Li, B. Y. *et al.* The role of FK506-binding proteins 12 and 12.6 in regulating cardiac function. *Pediatr Cardiol* **33**, 988-994, doi:10.1007/s00246-012-0298-4 (2012).
- 2 McCall, E. *et al.* Effects of FK-506 on contraction and Ca²⁺ transients in rat cardiac myocytes. *Circ Res* **79**, 1110-1121, doi:10.1161/01.res.79.6.1110 (1996).
- 3 Guo, T. *et al.* Kinetics of FKBP12.6 binding to ryanodine receptors in permeabilized cardiac myocytes and effects on Ca sparks. *Circ Res* **106**, 1743-1752, doi:10.1161/CIRCRESAHA.110.219816 (2010).
- 4 Gandon-Renard, M. *et al.* Dual effect of cardiac FKBP12.6 overexpression on excitation-contraction coupling and the incidence of ventricular arrhythmia depending on its expression level. *J Mol Cell Cardiol* **188**, 15-29, doi:10.1016/j.yjmcc.2024.01.003 (2024).
- 5 Jeyakumar, L. H. *et al.* FKBP binding characteristics of cardiac microsomes from diverse vertebrates. *Biochem Biophys Res Commun* **281**, 979-986, doi:10.1006/bbrc.2001.4444 (2001).
- 6 Zissimopoulos, S., Seifan, S., Maxwell, C., Williams, A. J. & Lai, F. A. Disparities in the association of the ryanodine receptor and the FK506-binding proteins in mammalian heart. *J Cell Sci* **125**, 1759-1769, doi:10.1242/jcs.098012 (2012).
- 7 Lam, E. *et al.* A novel FK506 binding protein can mediate the immunosuppressive effects of FK506 and is associated with the cardiac ryanodine receptor. *J Biol Chem* **270**, 26511-26522, doi:10.1074/jbc.270.44.26511 (1995).
- 8 Asghari, P. *et al.* Cardiac ryanodine receptor distribution is dynamic and changed by auxiliary proteins and post-translational modification. *Elife* **9**, doi:10.7554/eLife.51602 (2020).
- 9 Richardson, S. J., Thekkedam, C. G., Casarotto, M. G., Beard, N. A. & Dulhunty, A. F. FKBP12 binds to the cardiac ryanodine receptor with negative cooperativity: implications for heart muscle physiology in health and disease. *Philos Trans R Soc Lond B Biol Sci* **378**, 20220169, doi:10.1098/rstb.2022.0169 (2023).

We have included the following analysis and new Supplementary Fig. 6 in the calstabin result section: “Finally, it has been reported that calstabin-1, which is also expressed in the heart, binds and stabilizes RyR2 but with lower affinity than calstabin-2.^{51,52} We analyzed the structures of the complexes between RyRs and calstabins and determined that the identical structure and mode of

binding could explain the common stabilizing role that both calstabins have on RyR2, and that two negative residues (E32 and D33 in calstabin-1 vs Q32 and N33 in calstabin-2) could explain the lower affinity of RyR2 for calstabin-1 due to electrostatic repulsion (Supplementary Fig. 6).”

6. How does Rycal binding to RyR2 stabilize the RyR2 structure? No explanation is provided. A detailed structure-activity analysis would be needed to support the claim that Rycal binding stabilizes the closed state in the various CPVT mutant channels.

The explanation of how Rycal binds to and stabilizes the RyR structure was addressed in our previous publications Miotto et al 2022 and Melville et al 2022. We now have included this statement in the result section explaining how Rycal binding stabilizes the closed state in the various CPVT mutant channels:

“Similarly to RyR2-R2474S,¹⁰ we detected an increased density in the cleft of the RY1&2 domain of RyR2-R420Q, RyR2-R420W treated with ARM210, where Rycals were shown to bind (Supplementary Fig. 8e,f).^{10,54} We also detected an increased density in between the BSol1 and RY1&2 domains. This suggests that binding of ARM210 to the RY1&2 domain affects the local structure and stabilizes the interaction with the BSol1 domain, which in turn stabilizes the upward-inward conformation of the cytoplasmic shell associated with a stable closed pore of the channel in agreement with our previous reports (Supplementary Fig. 8e-i).^{10,54}”

7. The authors simulated the HF condition by incubating HEK cells in H₂O₂, resulting in reversible and irreversible oxidative modifications of RyR2. What molecular changes are induced is not examined. As such, there are multiple unknown modifications of the RyR2 channel that contribute to the altered CryoEM structure. Whether those modifications are prevented by Rycal treatment is not examined.

The post-translational modifications that affect the structure of RyR2 are unresolvable due to limitations in cryo-EM resolution. We have included this statement in the results section: “We were unable to detect any additional density in the cryo-EM map of RyR2-S2808D that could explain the incorporation of irreversible oxidative modifications due to the small size of the oxidative modifications (usually one oxygen atom), which is below the resolution limit achieved by cryo-EM.”

Secondly, we add ARM210 to the final purified RyR2 sample and not to the HEK cells, where the H₂O₂ is added, removing any possibility of prevention of post-translational modifications. Moreover, we have published extensively on this question reporting that Rycals DO NOT change the post-translational modifications.¹⁰⁻¹⁴ That is not the mechanism of action for this class of drugs. We have included this statement in the discussion section: “Treatment with Rycals reduces RyR open probability and ER/SR Ca²⁺ leak in diseased organs/tissues expressing the channel.⁷⁵ ...In the present study, we show that the beneficial effects of Rycals in HF are achieved by reversing the primed state and diminishing the RyR2-linked diastolic SR Ca²⁺ leak, and not by preventing post-translational modifications, as previously shown.^{16,18,76-78}”

Response References:

10 Dridi, H. *et al.* Heart failure-induced cognitive dysfunction is mediated by intracellular Ca(2+) leak through ryanodine receptor type 2. *Nat Neurosci*, doi:10.1038/s41593-023-01377-6 (2023).

- 11 Dridi, H. *et al.* Ryanodine receptor remodeling in cardiomyopathy and muscular dystrophy caused by lamin A/C gene mutation. *Hum Mol Genet* **29**, 3919-3934, doi:10.1093/hmg/ddaa278 (2021).
- 12 Shan, J. *et al.* Role of chronic ryanodine receptor phosphorylation in heart failure and beta-adrenergic receptor blockade in mice. *J Clin Invest* **120**, 4375-4387, doi:10.1172/JCI37649 (2010).
- 13 Mohamed, B. A. *et al.* Sarcoplasmic reticulum calcium leak contributes to arrhythmia but not to heart failure progression. *Sci Transl Med* **10**, doi:10.1126/scitranslmed.aan0724 (2018).
- 14 Cazorla, O. *et al.* Stabilizing Ryanodine Receptors Improves Left Ventricular Function in Juvenile Dogs With Duchenne Muscular Dystrophy. *J Am Coll Cardiol* **78**, 2439-2453, doi:10.1016/j.jacc.2021.10.014 (2021).

8. The structural data with the two CPVT variants and their effect on CaM binding are well-documented and could be reported as a stand-alone MS.

We believe that including all these data together allow us to propose a unifying mechanism for cardiac arrhythmias, as well as to further progress in the understanding of the regulation of RyR gating by the cytoplasmic shell (see section “Understanding the regulation of RyR gating by the cytoplasmic shell”). We have included this sentence in the abstract to make sure this is a clear take-home-message for the readers: “Here, we solved the cryogenic electron microscopy structures of ryanodine receptor 2 variants linked either to heart failure or inherited sudden cardiac death. All are in the primed state, part way between closed and open. Binding of Rycal drugs to ryanodine receptor 2 channels reverts the primed state back towards the closed state, decreasing Ca²⁺ leak, improving cardiac function, and preventing arrhythmias. We propose a structural-physiological mechanism whereby the ryanodine receptor 2 channel primed state underlies the arrhythmias in heart failure and arrhythmogenic disorders.”

And in the discussion: “To better understand the mechanistic underpinnings that result in leaky RyR2 channels that play a role in heart failure and arrhythmogenic disorders, we analyzed the structures of the calstabin-depleted phosphomimetic RyR2-S2808D channel and two CPVT-linked mutant channel structures, RyR2-R420Q and RyR2-R420W. Each of these structures show that the modified channels are in the primed state, defined as an intermediate conformation of the cytoplasmic shell between the closed and open WT RyR2 channels. Moreover, we determined that the conformation of the cytoplasmic shell can be affected by mutations, post-translational modifications, and ligand/protein binding, suggesting it plays a role in the regulation of the pore stability.”

Reviewer #2 (Remarks to the Author):

The RYR2 mutants, S2808D, R420Q and R420W, are closely associated with cardiomyopathy and CPVT, two different types of heart disease. The authors here reported cryo-EM structures of these RYR2 mutants at high resolution. Through comparing the structures of these RYR2 mutants with that of RYR WT, the authors observed some structural differences in the cytoplasmic domains of RYR2. The authors further claimed that the mutations shift the RYR2 to the primed state that is easier to be activated. This explains how the mutations cause the unwanted leaking of the

channel. Collectively, this work provides some molecular clues for how these mutations in RYR2 can cause heart disease. Overall, this is interesting work. The cryo-EM maps are of high quality. However, there are some concerns that need to be addressed. Here are my specific points:

(1) The authors compared the structure of RYR2 S2808D with previously determined RYR2 WT in the closed state. The RYR2 S2808D was purified from the cells under H₂O₂ treatment. If I understand correctly, RYR2 WT was purified from the cells without any H₂O₂ treatment. If this is the case, the structural comparison shown in Figure 1A and 1B would not be rigorous. We don't know for certain whether the detected conformational difference is caused by mutation or H₂O₂ treatment.

The reviewer is correct that we are comparing the RyR2-S2808D expressed in cells treated with H₂O₂ with dephosphorylated RyR2 expressed in cells without the treatment. As mentioned in the manuscript, our goal was to compare the control WT RyR2 to the heart failure RyR2, which exhibits PKA phosphorylation of residue RyR2-S2808 and oxidative modifications. Moreover, by comparing PKA phosphorylated WT RyR2 to RyR2-S2808D, we can conclude based on the similar RMSD that the PKA phosphorylation is the dominant factor in the conformational changes.

Similarly, the author mentioned that the cryo-EM structures of RYR2 R420Q/W were determined under conditions that simulate diastole. If the WT structure was not determined at the same condition, the comparison is not so meaningful.

We apologize for the lack of clarity in this section. The conditions that simulate diastole are ATP 10 mM and free calcium 150 nM, which are the same used for all datasets including the control dephosphorylated WT RyR2. We have corrected the methods where the information was missing.

(2) Based on Figure 1 and Figure 5, the peripheral domains undergo the largest movement between the WT and mutant RYR2. Nevertheless, the peripheral domains in the cryo-EM map of RYR2 mutants were resolved at relatively low resolution, indicating that these regions are very dynamic. It is not so meaningful to compare the conformation of a flexible domain between WT and mutant RYR2, because the resolved conformation is the average of many different conformational states. The authors need to perform 3D Variability Analysis with cryoSPARC to convince the readers that the observed conformational differences are not simply due to the intrinsic flexibility of the peripheral domains.

The reviewer has a point that the most outer domains (the RY1&2, the RY3&4 and the BSol2 domains) have lower resolution due to their intrinsic dynamic nature and could affect the atomic model and introduce noise or increase the uncertainty of the RMSD calculations. To generate more precise RMSD values, we reanalyzed the data including only those domains with best resolution: NTD, SPRY, JSol, and BSol1 (residues 1-861,1077-2681). We revised the method section to reflect this change: "The analysis performed here aims to quantify the relative position of the cytoplasmic shell (using the residues 1-861 and 1077-2681 that correspond to the well resolved NTD, SPRY, JSol, and BSol1 domains, see local resolution on Supplementary Fig. 14) of an atomic model X from the closed (PDB: 7UA5) and open (PDB:7UA9) models of WT RyR2."

We remade the figures using the reanalyzed RMSD values, which differ slightly from the original values without changing the overall interpretation.

As requested, we included the figure Supplementary Fig. 13 showing the 3D Variability Analysis with CryoSPARC demonstrating that the main mode of motion of the cytoplasmic shell with a normal distribution is similar among variants and conditions. This supports the use of the average map for building the atomic model. We have included this statement in the methods section:

“Since the RyR2 cytoplasmic shell is dynamic and shows a continuous range of conformations with a normal distribution around the average (Supplementary Fig. 13 and Supplementary Movie 3,4), the classes of good particles, which have only small differences in the cytoplasmic shell upward-inward or downward-outward position, were pooled together in order to work with the global average and not discrete classes randomly separated by the software.”

The authors need to show the local resolution map in the supplementary figures.

We had already included the resolution for each local refinement maps in the figures S2A-F. As requested, we now have added the Supplementary Fig. 14 showing the local resolution map. We have included this statement in the methods section. “The local resolution map of RyR2-S2808D, which is representative of all structures, is shown (Supplementary Fig. 14).”

(3) As shown in Figure. 3, the authors observed certain movement of NTD-B domain in relative to NSol domain, between WT and mutant RYR. However, the observed conformational change might not be significant, as the movement is too small (0.7 and 0.4 angstrom). It could be simply result from the uncertainty in model building due to the limited resolution (~3 angstrom). The author mentioned that this conformational change is supported by the previous MD simulation result, but this claim lacks support by any figures. This part needs to be elaborated. Ideally, the authors need to perform the MD simulation by themselves to validate the conformational difference observed in the cryo-EM structures.

We apologize for the lack of clarity in this section. The movement of 0.7 and 0.4 Å is of the 298-304 loop next to the residue 420 (figure 3d-f), not of the overall movement of the domain (figure 3a-c) which shows differences greater than 2Å in some parts. To make it clearer we have added the rotation degree of the domains and the distance shifted of the outer parts to Figure 3b,c. The global rotation of the NTD-B domain is observed in crystallography and MD simulations, not the movement of the internal loop. We initially included the comparison with the crystallographic and MD data as anecdotal, since cryo-EM of the full-length protein is more powerful than crystallographic and MD experiments performed on the isolated N-terminal domain. However, after careful analysis we have decided to eliminate this comparison because it does not add valuable information to the overall message.

As a commentary, the resolution achieved (~3Å) is not the uncertainty in model building. As shown in figure 3d-f, the cryo-EM maps have a resolution sufficient to see the backbone trace and the sidechains. For example, in figure 3f, the residue W420 has a very clear density that leaves very little uncertainty while building the model. We propose that the uncertainty in our comparisons can be measured as the RMSD of aligned domains that should be identical. In this case, when measuring the RMSD of the NSol between closed RyR2 (7U9Q) and primed RyR2-

R420Q (8UXC) or primed RyR2-R420W (8UXF) the values are 0.37 Å and 0.34 Å, respectively. While 0.7 Å is above the uncertainty level, 0.4 Å is close to it. We have clarified this in the results section.

“However, due to its larger size and steric effect, the R420W mutation results in a 0.4 Å downward movement of the 298-304 loop, which might be not significant as it is close to the uncertainty level, measured as the RMSD between aligned NSol domains (0.34 Å), causing a less pronounced but significant rotation of the NTD-B compared to RyR2-R420Q (Figure 3f).”

(4) During the image processing of each dataset, the authors merge all the classes that represent the closed state together. It is unclear to me why these classes can be combined. Do they have exactly the same conformation? Do the peripheral regions also adopt the same conformation? If not, the authors need to analyze each class individually.

We apologize for the lack of clarity in this section. The cytoplasmic shell of RyRs has a continuum of conformations that the software will separate into classes if imposed. However, the different conformations have a normal distribution around the average (Supplementary Fig. 13) and therefore it is better to work with the average knowing that the cytoplasmic shell is still dynamic. We have expanded the methods section and included new figures and movies:

“A second round of heterogeneous refinement with four classes including “RyR-like” volumes was performed to further separate “good” particles from “bad” particles (partially misfolded particles that arise from the purification process or due to the interaction with the air-water interface). Since the RyR2 cytoplasmic shell is dynamic and shows a continuous range of conformations with a normal distribution around the average (Supplementary Fig. 13 and Supplementary Movie 3,4), the classes of good particles, which have only small differences in the cytoplasmic shell upward-inward or downward-outward position, were pooled together in order to work with the global average and not discrete classes randomly separated by the software.”

Reviewer #3 (Remarks to the Author):

In this manuscript cryo-EM was used to compare the structures of cardiac isoform of the ryanodine receptor (RyR2) with various variants of RyR2 linked to heart failure. The authors found that mutations in RyR2 associated with heart failure make the channel more susceptible to a leak of calcium out of the sarcoplasmic reticulum. The leak occurs because the protein is placed in a primed state, between closed and open states, and binding of calstabin-2, a protein that stabilizes the closed RyR2, is reduced. This leak leads to afterdepolarizations, ventricular tachycardia, and sudden cardiac death. The authors also found that treatment with a Rycal drug helps maintain the channel in the closed state, which decreases the calcium leak and increases calstabin-2 binding.

Overall, this manuscript provides a very detailed analysis of RyR2 structures. As stated on line 411 these structures are consistent with previously published structures and similar conclusions were reached. The comparison between wild type RyR2 and several versions with pathogenic mutations provides ample justification for the selection of the mutations studied. The basic findings, that the mutations make RyR2 more leaky and that a Rycal drug can compensate for the altered structural changes, have been presented in previous work by this group. The specific

mutations associated with known cardiac diseases makes the findings of particular interest to cardiologists, especially those developing treatments.

Line 161: explanation of the effects of rapamycin is needed.

Explanation added: “Rapamycin is a macrocyclic drug that specifically binds to calstabins in the same site as RyRs do, sequestering calstabins and preventing their interaction with RyRs. Therefore, it prevents calstabin binding to RyRs.”

Line 305: the observation that “the magnitude of structural changes induced by the CPVT-linked mutants is associated with the level of pathogenicity of the disease” provides strong support for the suggestions made by the authors. This finding should be further highlighted.

We appreciate this suggestion. We further highlighted this finding in the discussion. “We found that the differences in the level of the downward-outward movement of the cytoplasmic shell correlate with the age of onset of CPVT patients, further supporting the importance of the primed state in the development of these pathologies.”

Line 333: the systolic calcium concentration is stated as 40 μM , but this seems too high if mimicking cytoplasmic calcium levels and too low for intra-SR calcium levels. More explanation for the selection of 40 μM should be provided.

We have included the explanation of the choice of concentration in the results section. “We solved the cryo-EM structure of the RyR2-R420W channels under conditions that resemble the Ca^{2+} -induced Ca^{2+} release (CICR) that occurs during systole (Supplementary Fig. 2e,f, 3c). At the beginning of the systole, when CICR is the highest, the local Ca^{2+} concentration in the dyadic cleft reaches 10-100 μM or higher.^{64,65} Here, we used 40 μM because this is the concentration at which ryanodine binding and lipid bilayer experiments result in maximum open probability, preventing RyR2 from reaching the closed or inactivated states.”

Line 492: The authors assert that the “channel population with mixed WT:CPVT protomers (possible combination of protomers WT:CPVT would be 4:0, 3:1, 2:2, 1:3, and 0:4), with only 20% of the channels having all four mutant protomers.” Unless the distribution was measured, the authors cannot assert that 20% of the channels would have only mutant protomers. The likelihood of preferential classes is high.

We agree with the reviewer that there is no experimental measurement and we removed this from the discussion.

The object labeled “C” in Fig 7 and the intro figure needs to be defined in the legend.

We apologize for the lack of clarity. The object “C” is calstabin. We changed the figure for better understanding and include it in the legend.

Throughout the manuscript the authors use the phrase “suggesting it might” (or something similar). This phrase weakens the comment. Either the idea “suggests” the concept or “it might” influence the concept.

We thank the reviewer for this insight. We have corrected the phrase when used.

REVIEWER COMMENTS

Reviewer #2 (Remarks to the Author):

The manuscript has been improved significantly after the revision. However, I am still not unconvinced about using the RMSD value to characterize the conformational variance among different datasets. I would suggest the authors to calculate and compare the conformational landscape of different datasets using other well-established method, such as manifold-based analysis, CryoDRGN or e2gmm in EMAN2. As mentioned by the authors, the manifold-based analysis has been used in the conformational analysis of RYR1. Such analysis will fully convince the readers that the observed conformational differences are significant.

Reviewer #3 (Remarks to the Author):

The authors have made reasonable changes to the comments.

Please check for typos – for example is it TCEP or TECP? Also define at first appearance.

Reviewer #4 (Remarks to the Author):

The manuscript by Miotto et al. adds significant amount of new structural data on mutant RyR2 ion channel. The presented data provides novel insights on RyR2 structural changes caused by CPVT-associated mutation as well as to the heart failure-associated biophysical mutation (S2808D). The revised manuscript adequately responds to the reviewers' concerns by providing massive structural data. The MS, I believe, provides a unifying hypothesis for the pathology of CPVT and Heart Failure and adds significantly to our understanding of structure/function of RyR2 and the pathology associated with the presented mutations.

My only concern is about the purification procedure of the recombinant mutant RyR2s.

Authors have purified the mutant RyR2 protein based on high affinity binding between calstabin2 and RyR2. In the past two decades, however, these authors have extensively demonstrated that PKA phosphorylation of RyR2 including its S2808D phospho-mimicking and CPVT mutations, dissociate calstabin2 from RyR2 rendering the channel leaky. Considering these previous reports, the very leaky RyR2 mutants may have been excluded during the purification procedure from the cumulated data. Is it possible to estimate the fraction of leaky cell population in these samples?

Reviewer's Comments:

Reviewer #2 (Remarks to the Author)

The manuscript has been improved significantly after the revision. However, I am still not unconvinced about using the RMSD value to characterize the conformational variance among different datasets. I would suggest the authors to calculate and compare the conformational landscape of different datasets using other well-established method, such as manifold-based analysis, CryoDRGN or e2gmm in EMAN2. As mentioned by the authors, the manifold-based analysis has been used in the conformational analysis of RYR1. Such analysis will fully convince the readers that the observed conformational differences are significant.

Our response:

We thank the reviewer for the positive comments. Using distance (RMSD values) and flexion angle measurements have been the standard tools for analyzing conformational changes in the Ryanodine Receptor field, especially when comparing mutants and different conditions (see below). We prefer RMSD over measuring single distances or angles, because RMSD analysis considers the changes of all residues while single distance or angle measurements might overlook unconventional conformational changes. The current RMSD analysis is an upgraded version of what we already published in our previous paper: Miotto et al. "Structural analyses of human ryanodine receptor type 2 channels reveal the mechanisms for sudden cardiac death and treatment" *Science Advances* (2022)

Most recent publications use distance or RMSD measurements to analyze conformational changes of RyRs:

- Haji-Ghassemi et al. Cryo-EM analysis of scorpion toxin binding to Ryanodine Receptors reveals subconductance that is abolished by PKA phosphorylation *Science Advances* (2023)
- Cholak et al. Allosteric modulation of ryanodine receptor RyR1 by nucleotide derivatives *Structure* (2023)
- Iyer et al. Molecular mechanism of the severe MH/CCD mutation Y522S in skeletal ryanodine receptor (RyR1) by cryo-EM *PNAS* (2022)
- Nayak et al. Ca²⁺ inactivation of the mammalian ryanodine receptor type 1 in a lipidic environment revealed by cryo-EM *eLife* (2022)
- Woll et al. Pathological conformations of disease mutant Ryanodine Receptors revealed by cryo-EM *Nature Communications* (2021)
- Iyer et al. Structural mechanism of two gain-of-function cardiac and skeletal RyR mutations at an equivalent site by cryo-EM *Science Advances* (2020)
- Ma et al. Structural basis for diamide modulation of ryanodine receptor *Nature Chemical Biology* (2020)
- Des Georges et al. Structural Basis for Gating and Activation of RyR1 *Cell* (2016)

ManifoldEM was developed and first used in RyR1 by our close colleagues and collaborators (Dashti et al. Retrieving functional pathways of biomolecules from single-particle snapshots *Nature Communications* (2020)). As with cryoDRGN and e2gmm, manifoldEM generates a conformational landscape from cryo-EM particles but presents two challenges: the axes of the landscape (a.k.a. conformational coordinates or UMAP) are generated for each processing or dataset to be analyzed. So, to be comparable among each other, all of our 10 datasets should be processed simultaneously, something we haven't seen done in the literature and that might present technical challenges. Although the conformational landscapes potentially achieved by these methods might describe the conformational space of our RyR2 channels, it might not improve the manuscript considering the scope of our analyses and message. We believe that our RMSD analyses, which are the norm in the field, have the advantage of being understandable by a broad audience: values close to 0 are more benign, values close to 1 are more pathogenic.

Reviewer #3 (Remarks to the Author)

The authors have made reasonable changes to the comments. Please check for typos – for example is it TCEP or TECP? Also define at first appearance.

Our response:

We appreciate the positive comments of the reviewer. The author is correct, it is TCEP (tris(2-carboxyethyl)phosphine). We will define it at first appearance.

Reviewer #4 (Remarks to the Author):

The manuscript by Miotto et al. adds significant amount of new structural data on mutant RyR2 ion channel. The presented data provides novel insights on RyR2 structural changes caused by CPVT-associated mutation as well as to the heart failure-associated biophysical mutation (S2808D). The revised manuscript adequately responds to the reviewers' concerns by providing massive structural data. The MS, I believe, provides a unifying hypothesis for the pathology of CPVT and Heart Failure and adds significantly to our understanding of structure/function of RyR2 and the pathology associated with the presented mutations. My only concern is about the purification procedure of the recombinant mutant RyR2s. Authors have purified the mutant RyR2 protein based on high affinity binding between calstabin2 and RyR2. In the past two decades, however, these authors have extensively demonstrated that PKA phosphorylation of RyR2 including its S2808D phospho-mimicking and CPVT mutations, dissociate calstabin2 from RyR2 rendering the channel leaky. Considering these previous reports, the very leaky RyR2 mutants may have been excluded during the purification procedure from the cumulated data. Is it possible to estimate the fraction of leaky cell population in these samples?

Our response:

We appreciate the quick response and positive comments of the reviewer. Regarding the concern about excluding the very leaky RyR2 channels due to weaker calstabin-2 binding (binding is necessary for the purification step involving the glutathione-matrix column or GSTrap), we estimate that a minimal fraction of RyR2 has been excluded or lost during the purification, and, therefore, all populations of RyR2 were included in our cryo-EM analysis. This is evidenced as no detectable band in the flow-through lane of the GSTrap purification step, compared to the input lane of the same step, as seen in the SDS-PAGE for RyR2-S2808D (supplementary figure 1g, see red box below) and western blots for RyR2-R420Q and RyR2-R420W (supplementary figure 7a,c, see red box below). If a significant population of very leaky RyR2 was not bound to the GSTrap column, it would show as a band in the flow-through lane. The absence of a detectable band in the flow-through indicates that most RyR2, if not all, is bound to the GSTrap matrix. We can explain binding of very leaky RyR2 channels to the GSTrap, since for the purification procedure, we use an excess of the GST-calstabin-2 fusion protein compared to RyR2. Also, GST-calstabin-2 binds to the GSTrap matrix which prevents diffusion. Both the excess concentration and the fixed calstabin-2 would result in an extraordinarily high local concentration of calstabin-2 that would increase the k_{on} and would shift the equilibrium towards the RyR2:calstabin-2 complex formation explaining the purification of all RyR2 particles in solution, as seen in the gel/blots.

Cropped from supplementary figure 1g (brightness and contrast were increased to better display the bands). The red box encases the RyR2 band in the GSTrap input lane, and the absence of the band in the GSTrap flow-through lane.

Cropped from supplementary figure 7a,c. The red box encases the RyR2 band in the GSTrap input lane, and the absence of the band in the GSTrap flow-through lane.

a PKA phosphorylated hRyR2-R420Q

c PKA phosphorylated hRyR2-R420W